



# 1 Evaluation of a low-cost optical particle counter (Alphasense OPC-N2)
# 2 for ambient air monitoring

Leigh R. Crilley[1], Marvin Shaw[2], Ryan Pound[2], Louisa J. Kramer[1], Robin Price[3], Stuart
Young[2], Alastair C Lewis[2], Francis D. Pope[1*]
[1]*School of Geography, Earth and Environmental Sciences, University of Birmingham,*
*Birmingham, United Kingdom, B15 2TT.*
[2]*National Centre for Atmospheric Science, Wolfson Atmospheric Chemistry Laboratories,*
*University of York, York, United Kingdom, YO10 5DD.*
[3]*Birmingham Open Media (BOM), 1 Dudley Street, Birmingham, B5 4EG.*
*Corresponding author – f.pope@bham.ac.uk

## 13 Abstract

A fast growing area of research is the development of low-cost sensors for measuring air
pollutants. The affordability and size of low-cost particle sensors makes them an attractive
option for use in experiments requiring a number of instruments such as high density spatial
mapping. However, for these low-cost sensors to be useful for these types of studies their
accuracy and precision needs to be quantified. We evaluated the Alphasense OPC-N2, a
promising low-cost miniature optical particle counter, for monitoring ambient airborne
particles at typical urban background sites in the UK. The precision of the OPC-N2 was
assessed by co-locating 14 instruments at a site to investigate the variation in measured
concentrations. Comparison to two different reference optical particle counters as well as a
TEOM-FDMS enabled the accuracy of the OPC-N2 to be evaluated. Comparison of the OPC-
N2 to the reference optical instruments demonstrated reasonable agreement for the measured
mass concentrations of $PM_1$, $PM_{2.5}$ and $PM_{10.}$ However, the OPC-N2 demonstrated a
significant positive artefact in measured particle mass during times of high ambient RH
(>85%) and a calibration factor was developed based upon κ-Kohler theory, using average
bulk particle aerosol hygroscopicity. Application of this RH correction factor resulted in the
OPC-N2 measurements being within 33% of the TEOM-FDMS, comparable to the agreement
between a reference optical particle counter and the TEOM-FDMS (20%). Reasonable inter-
unit precision for the 14 OPC-N2 sensors was observed. Overall, the OPC-N2 was found to





accurately measure ambient airborne particle mass concentration provided they are i)
correctly calibrated and ii) corrected for ambient RH. The reasonable level of precision
demonstrated between multiple OPC-N2 suggests that they would be suitable device for
applications where the spatial variability in particle concentration was to be determined.

## 1.0 Introduction

Airborne particles are of global concern due to their detrimental health effects, particularly in
the fine fraction ($PM_{2.5}$, particles with an aerodynamic diameter less than 2.5 µm) and as a
result are a regulated pollutant in the EU, USA and other states. Monitoring ambient particle
mass concentrations is typically performed using a small number of fixed instruments with
gaps in the spatial coverage usually estimated via modeling or interpolation. This is often
unsatisfactory as there can be micro-environments in urban areas that result in large spatial
and temporal inhomogeneity in airborne particle concentrations, which in turn makes
assessment of human exposure to airborne particles difficult (de Nazelle et al., 2017).
Into this gap a fast growing area is the development of low-cost sensors for measuring the
concentrations of a wide range of species in the atmosphere including gases and particles
(Lewis et al., 2016; Rai et al., 2017; Snyder et al., 2013). However the question remains as to
whether the uncertain quality of data from these low cost sensors can be of value when
attempting to determine pollutant concentrations at high spatial resolution (Kumar et al.,
2015). Sensors for both gases and particles can suffer from drift and a number of interference
artefacts such as relative humidity (RH), temperature and other gas phase species (Lewis et
al., 2016; Mueller et al., 2017; Popoola et al., 2016). Despite these challenges, recent work
has shown that low-cost gas sensors can be deployed in large scale networks provided
appropriate corrections for known artefacts are applied (Borrego et al., 2016; Mead et al.,
2013; Mueller et al., 2017), with clustering of multiple gas sensors into one unit shown to be
an effective methodology (Lewis et al., 2016; Mueller et al., 2017; Smith et al., 2017).
For low-cost particle sensors, their reported performance across the literature is somewhat
mixed (Borrego et al., 2016; Castellini et al., 2014; Sousan et al., 2016; Viana et al., 2015)
and can depend on the type of particle sensor employed. There are a wide range of low-cost
particle sensors are available commercially by companies including from manufacturers
Dylos, TSI, Airsense and Alphasense. The more widely used and available low-cost particle



sensors can be considered as miniaturized versions of optical particle counters (OPC) and
employ a light scattering technique to measure ambient particle concentrations (See e.g. (Gao
et al., 2015; Sousan et al., 2016). While these miniature OPC are not meant to compete with
more established instrumentation in terms of their accuracy and precision, their affordability
and size makes them attractive for use in experiments requiring a number of such
instruments, such as personal monitoring (See e.g. (de Nazelle et al., 2017; Steinle et al.,
2015)). However to be useful in these types of studies, the precision and accuracy of these
instruments needs to be evaluated.
Laboratory assessments of the performance of a number of low-cost miniature OPC's have
shown promising results, with reasonable precision observed compared to reference
instrumentation (Manikonda et al., 2016). Sousan et al., (2016) evaluated the Alphasense
OPC-N2 in a laboratory study using reference aerosols (Arizona road dust, NaCl and welding
fumes) and found reasonable agreement for size distributions and particle mass between the
OPC-N2 and a GRIMM Portable Aerosol Spectrophotometer, provided appropriate and
specific calibrations were applied. While these results are encouraging (Manikonda et al.,
2016; Sousan et al., 2016), laboratory-based studies using reference aerosols may not be
representative of their performance when measuring ambient particles, owing in part to the
complex mixture and variable relative humidity and temperature encountered in the real-
world. Previous field testing of low-cost particle sensors has found that that Dylos (Steinle et
al., 2015), PUWP (Gao et al., 2015) performed well for ambient sampling of particle mass
concentration in both an urban and rural environments when compared to reference
instruments however were assessed over a short period (4-5 days). In contrast, at a roadside
location poor agreement between two different OPC sensors compared to reference
instruments was observed by Borrego et al. (2016). Clearly, the results are mixed and longer-
term assessment of the stability and longevity of these instruments are needed, as these are
critical parameters when considering their worth for use in large-scale networks.
We evaluate here the Alphasense OPC-N2, a promising low-cost miniature optical particle
counter (Sousan et al., 2016), for monitoring ambient airborne particles at typical urban
background sites in the UK. We assessed the inter-unit precision of the OPC-N2 by co-
locating 14 instruments at a single site to investigate the variation in measured particle mass
concentration in the $PM_{10}$, $PM_{2.5}$ and $PM_1$ size fractions between OPC-N2. In order to
determine the accuracy of the OPC-N2, we compared it to two well-established commercial



optical particle counters that employ a similar light scattering technique as well as a TEOM-
FDMS, a regulatory standard instrument for particle mass concentration measurements.
**2.0 Method**
**2.1 Instrumentation**
**2.1.1 Alphasense Optical particle sensor (OPC-N2)**
The Optical Particle Sensor (OPC) under evaluation in the current work is the OPC-N2
manufactured commercially by Alphasense (www.alphasense.com). The OPC-N2 can be
considered as a miniaturized OPC as it measures 75x60x65 mm and weighs under 105 g, and
as such is significantly cheaper (approx. £200) than the comparable reference instruments
(see next section). The OPC-N2 has a reported size range of 0.38 to 17 μm across 16 size
bins, and maximum particle count of 10,000 per second. All OPC-N2 in this study were
firmware version 18.
The OPC-N2 is designed to log data via a laptop using software supplied by Alphasense,
however this may not be practical when using multiple OPC-N2 at once or for personal
monitoring. Therefore, we developed a custom built systems for logging the OPC-N2 during
the inter-comparison, using custom-built logger utilizing Raspberry Pi 3 and Arduino
systems. The Python code to log the outputs from OPC-N2 on a Raspberry Pi 3 is made
available in the Supplementary Material. The Python code makes use of the py-opc python
library for operating the OPC-N2 written by Hagan (2017).
**2.1.2 Reference Instruments**
The first reference instrument was a TSI 3330 optical particle spectrophotometer (OPS),
which measures particles number concentrations between 0.3 – 10 μm across 16 size bins,
with a maximum particle count of 3000 particles cm$^{-3}$. A GRIMM Portable Aerosol
Spectrometer (PAS-1.108, forthwith referred to as the GRIMM) was also utilized, which
records particle number concentrations in 15 bins from 0.3 – 20 μm. The TSI 3330 and
GRIMM were both recently calibrated and serviced. All measurements of airborne particle
concentrations are inherently operationally defined and as a result the TSI 3330 and the
GRIMM were chosen as reference instruments as they measure particle size in similar size
bins by a similar photometric technique to the Alphasense OPC-N2.


For the sake of this inter-comparison, we have taken the TSI 3330 and GRIMM data as an
accurate measure of particle mass concentrations. The reference instrument used for the
factory calibration of the OPC-N2 by Alphasense is the TSI 3330 (Sousan et al., 2016) and
hence included for comparison.

## 2.2 Inter-comparison locations

### 2.2.1 Elms Rd Observatory Station

The instruments were housed within the Elms Road Observatory Station (EROS) located on
the University of Birmingham campus. The site is classed as urban background, with
emissions from nearby road and a construction site the major sources of particles. Fourteen
OPC-N2 were deployed at EROS, enabling the precision of the OPC-N2 to be assessed along
with the accuracy relative to the reference instruments, the TSI 3330 and GRIMM. An
intensive inter-comparison ran for just over 5 weeks, from $26^{th}$ August till $3^{rd}$ October 2016,
during which all 14 OPC-N2, TSI 3330 and GRIMM sampled ambient air. Minimal lengths
of stainless steel tubing (OPC-N2) and conductive black tubing (TSI 3330 and GRIMM)
were used to sample outside air, with each OPC having its own inlet at a height of 1.5 m.
Sampling intervals for the OPC-N2, TSI 3330 and GRIMM were 10, 60 and 6 seconds,
respectively. In addition, measurements from the nearby Elms Road Meteorological station
were also obtained which is located approximately 100 m away from EROS.
At the conclusion of the intensive inter-comparison, a subset of the OPC-N2 (5) continued to
sample at EROS along with the GRIMM, to test the robustness and suitability of the OPC-N2
for longer-term monitoring. The long-term monitoring concluded on 1 February 2017,
meaning that these OPC-N2 sampled ambient air for up to 5 months.

### 2.2.2 Tyburn Rd

For regulatory purposes, an accepted method for measuring particle mass concentrations is a
Tapered Element Oscilating Microbalance (TEOM) and therefore we also compared the
OPC-N2 to this technique despite the difference in particle measurement approaches. An
urban background air monitoring station part of the UK Automatic and Rural Urban Network
(AURN) nearby EROS (Tyburn Rd) was chosen for this inter-comparison. At the Tyburn Rd
AURN station, the TEOM monitor was fitted with a Filter Dynamic Measurement System
(FDMS) (Grover et al., 2006). A subset of OPC-N2 (4) and the GRIMM PAS 1.108 that were





deployed at EROS sampled at Tyburn Rd station for 2 weeks during February 2017. The
OPC-N2 was housed individually within waterproof boxes on the roof of the cabin near to the
TEOM inlet in order to keep the inlet length the same as used at EROS. The GRIMM
sampled from a nearby separate inlet.
**2.3 Data Analysis**
All OPC employed in this study count the number of particles and determine the size based
upon particle light scattering of a laser, and to convert to particle mass concentration must
apply a number of assumptions. To calculate the particle mass concentration, spherical
particles of a uniform density and shape are assumed, which is not strictly true for airborne
particles in an urban atmosphere but is considered a reasonable approximation. Therefore to
ensure a fair comparison between the different OPC, the same calculations and assumptions
must be applied to all three OPC measurements. The TSI 3330 data was processed using the
TSI AIM software to convert the particle count concentration to particle mass measurements.
The particle counts from the GRIMM data was converted to particle mass (via particle
volume) using the same calculations, as outlined in the TSI AIM software manual according
to Equations 1 to 3:
$$D_{pv} = LB\left[\frac{1}{4}\left(1 + \left(\frac{UB}{LB}\right)^2\right)\left(1 + \left(\frac{UB}{LB}\right)\right)\right]^{\frac{1}{3}}$$    (1)
$$v = \frac{\pi D_{pv}{}^3 n}{6}$$    (2)
$$m = \rho v$$    (3)
where $D_{pv}$ is the volume weighted diameter, LB the channel lower boundary, UB the channel
upper boundary, $v$ is the particle volume for a channel, n is number weighted concentration
per channel, m is the particle mass per channel and ρ is the particle density.
The OPC-N2 converts, on board via a factory determined calibration, particle counts to
particle mass concentration in $PM_1$, $PM_{2.5}$ and $PM_{10}$ mass concentrations. There is no further
information provided by Alphasense on how this calculation is performed apart from the
applied particle density across all size bins was 1.65 g cm$^{-3}$. Therefore, we assumed





calculations are similar to Eqns 1 and 2 as applied to the TSI and GRIMM data and used the
same particle density (1.65) across all size bins to calculate particle mass for all OPC.
All instrument time series were corrected for drift against a reference time. As the sampling
intervals varied slightly between the different OPC, a 5 min average of particle
concentrations was used for inter-comparison between instruments.

## 3.0 Results and Discussion

### 3.1 EROS inter-comparison

### 3.1.1 Comparison of reference optical light scattering instruments

The two light scattering optical particle counters used as reference instruments in this study
were found to be  well correlated ($r^2 > 0.9$), with the GRIMM recording between 20-30%
higher concentrations for all three particle mass fractions (Fig S1, Supporting Information).
The GRIMM is known to overestimate number concentration (Sousan et al., 2016 and
references therein) and this difference may reflect differing efficiencies in particle detection
between the two instruments.

### 3.1.2 Performance of the OPC-N2

The performance of the custom built logging systems varied between 44-94% successful data
capture, with the Arduino and Raspberry Pi systems giving 44-65% and >92%, respectively.
The Raspberry Pi data logger system was used for the long-term measurements and for the
inter-comparison with the AURN site due to its better performance. The data losses were due
to hardware issues and not related to performance of the OPC-N2. Due to the missing data,
only a subset of measured $PM_{2.5}$ concentrations when all 14 OPC-N2 were logging are shown
in Fig 1, along with measured concentrations by the reference instruments. From Fig 1, while
there are times when there appears to be excellent agreement between the OPC-N2 and the
reference instruments, there are times when the OPC-N2 record a significant positive artefact,
and during these times the spread in measured concentrations increases. For example, on the
morning of the 18th September, the range of measured concentrations by the individual OPC-
N2 was from approximately 30-150 µg m$^{-3}$, whereas the reference instruments reported ~10
µg m$^{-3}$. The cause of the positive artefact is investigated in later sections, but it points to the
individual OPC-N2 responding differently to this artefact. Similar trends were also observed
for $PM_1$ and $PM_{10}$, see Figure S2 in the Supporting Information.





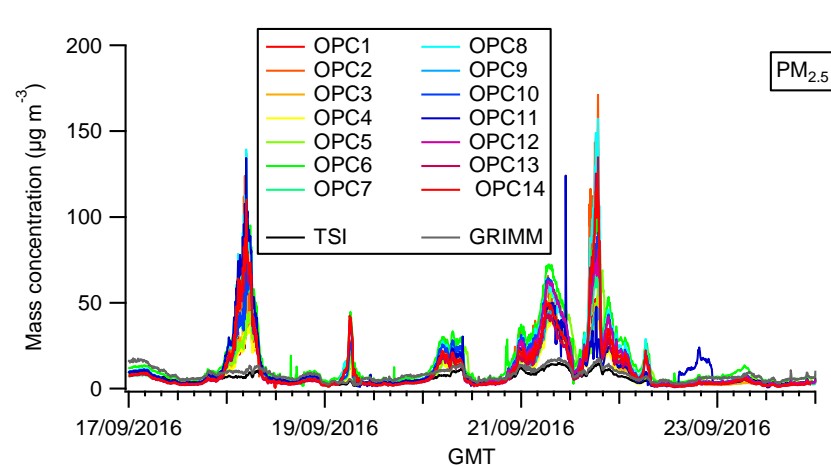

Figure 1: Time series of PM$_{2.5}$ concentrations measured by all OPC-N2 and the reference instruments, TSI 3330 and GRIMM for selected period with high OPC-N2 data coverage.

As there is a considerable spread in response for the OPC-N2 relative to the reference instruments, we then quantified whether it was always the same OPC-N2 reading low and high. Due to the aforementioned data capture issues, this analysis was only applied to days when all 14 OPC-N2 were running, 21$^{st}$-24$^{th}$ September (Fig 1). The results are shown as a rank order plot, where the OPC-N2 observations are ordered from the highest reported value to the lowest over this period, normalised to the median concentration at the start of the analysis (t=0), shown for PM$_{2.5}$ mass concentration in Figure 2. The ranking of the OPC-N2's showed some variability over time within periods of 1-6 hours, which was particularly noticeable during periods when the OPC-N2 signals underwent large changes in concentrations. This demonstrates that the highest reporting OPC was not consistently reporting the highest and lowest the lowest PM$_{2.5}$ concentration over the whole 3 day period. The same trend was also observed for PM$_1$ and PM$_{10}$ mass concentrations, as shown in Figure S3 (Supporting Information).

For the 3 day time period (21$^{st}$-24$^{th}$ of September) we applied the rank order analysis, two subsets of concentrations measured by the OPC-N2 were evident in the time series (Fig 1); one a period of highly variable mass concentrations (0:00 21/9/16 to 12:00 22/9/16) of September) followed by more stable mass concentrations (12:00 22/9/16 onward). This was reflected in the corresponding rank order plots where relatively consistent OPC rank orders were observed throughout the variable and comparatively stable PM concentrations periods.



However, there is a noticeable transition between the two periods in the rank order plot,
observed at approximately 12:00 on the 22$^{nd}$). This transition in rank orders would reflect the
difference in OPC PM sensitivities, random noise and offset values between each OPC. Over
the 3 day period the OPCs appeared to hold their response characteristics and hence rank
orders well, suggesting that over this timescale quantitative concentrations could be directly
compared. Due to the changing response and the incomplete data coverage, for the rest of the
analysis in this paper, when comparing to the reference instruments the median and inter-
quartiles concentrations of all 14 OPC-N2 were used.

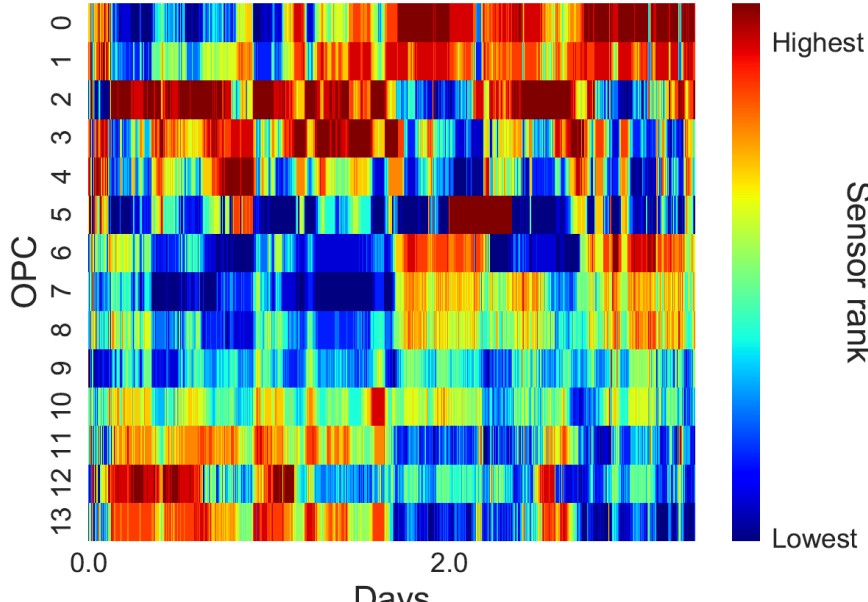

Figure 2: Sensor ranking analysis for measured $PM_{2.5}$ mass concentrations for the 14 OPC-
N2 over a 3 day period (21st-24th of September) with high OPC-N2 data coverage.
One measure of the precision of a group of instruments is the coefficient of variance (CV)
and this was calculated for the measured ambient mass concentrations of all 14 OPC-N2 to
assess the variability between 14 instruments. The average CV was 0.32+0.16, 0.25+0.14 and
0.22+0.13 for $PM_1$, $PM_{2.5}$ and $PM_{10}$ mass concentrations, respectively. This is higher than the
value of 0.1 considered acceptable for duplicate instruments by the US EPA (see Sousan et
al., 2016 and references therein) but perhaps not unreasonable for low-cost sensors. This may





in part be due the OPC-N2 all sampling from separate but identical inlets but suggests the
precision of the OPC-N2 would need to be considered when comparing multiple units. To
analyse whether the CV for the OPC-N2 varied over the month, the median concentration
was plotted along with the CV (shown for $PM_{2.5}$ in Fig 3). Throughout the measurement
period, the CV was fairly consistent, with spikes in CV values evident during periods of high
$PM_{2.5}$ concentrations, in agreement with trends observed in Fig 1. We observed a similar
trend of consistent CV values for both $PM_1$ and $PM_{10}$ concentrations suggesting reasonably
stable agreement between all OPC-N2 over a 5 week period.

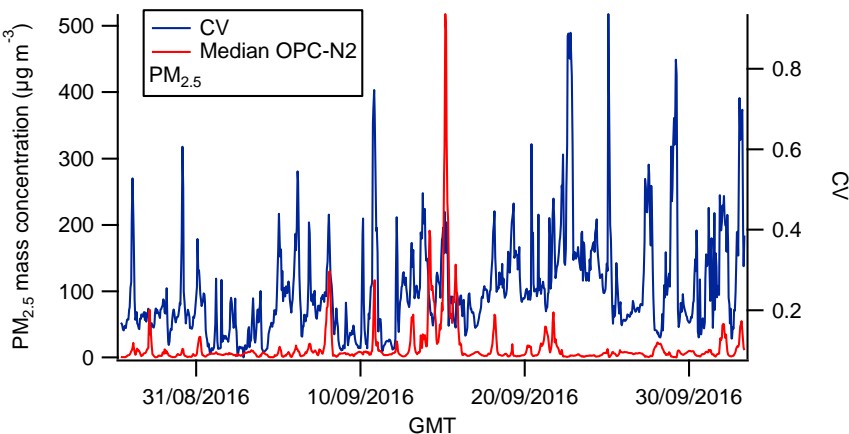

Figure 3: Time series of the hourly average median OPC and CV during the September
intensive inter-comparison at EROS for $PM_{2.5}$ mass concentration.
**3.2 Comparison of Alpha sense OPC to reference instruments**
**3.2.1 Particle mass concentration measurement at EROS**
The median and inter-quartiles of the measured PM concentrations from the 14 OPC-N2 were
used to compare the measured particle mass concentrations to the reference instruments
(Figure 4). From Fig 4, the notably similar distributions across all three particle size fractions
for the first and third quartiles indicate good agreement between the 14 OPC-N2, further
highlighting the reasonable degree of precision between the OPC-N2 as shown in the
previous section. At typical ambient $PM_{2.5}$ and $PM_{10}$ mass concentrations for the UK, similar
distributions were observed for the OPC-N2 and reference instruments (Fig 1), suggesting
reasonable agreement between the devices. In contrast, different distributions were observed




for the PM$_1$ fraction, with the OPC-N2 and GRIMM in agreement but appearing to over-
estimating the PM$_1$ mass concentrations with respect to the TSI 3330. While the OPC-N2 has
a higher particle size cut-off (0.38 μm) compared to the TSI (0.3μm) and may explain the
observed difference in frequency distribution for PM$_1$ (Fig 1), the TSI and GRIMM have the
same particle size cut-off (0.3 μm) and so would be expected to agree.

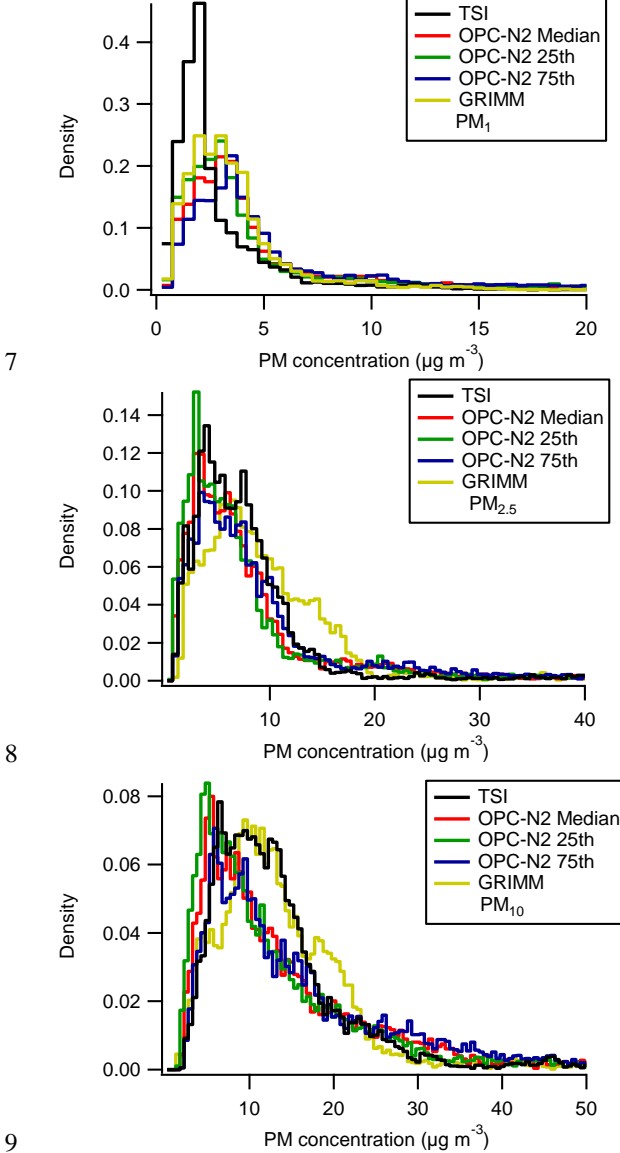





Figure 4: Histogram of measured $PM_1$, $PM_{2.5}$ and $PM_{10}$ mass concentrations by the TSI 3330,
GRIMM and median and inter-quartile values for the 14 OPC-N2. Note the different x and y
axis scales.
When the median and inter-quartile OPC-N2 concentrations were plotted against the TSI and
GRIMM concentrations, the slope was greater than unity for all three size fractions (Table 1)
indicating that the OPC-N2 were over-estimating the ambient particle mass concentrations
(approx. 2 to 5 times, Table 1). Overall, the OPC-N2 and GRIMM were in better agreement
compared to the TSI for all size fractions (Table 1). The GRIMM was found to record PM
concentrations 20-30% higher compared to the TSI (Figure S1), and this could in part
account for the observed lower slopes between the GRIMM and the OPC-N2.
Table 1: Slopes of measured PM mass concentrations of the reference instruments against the
median and inter-quartiles for OPC-N2. Correlation co-efficient, $r^2$ is given in parenthesis.

| OPC-N2 | PM$_1$ | | PM$_{2.5}$ | | PM$_{10}$ | |
|---|---|---|---|---|---|---|
| | TSI | GRIMM | TSI | GRIMM | TSI | GRIMM |
| **25th** | 2.93+0.01 (0.9) | 2.34+0.1 (0.92) | 3.16+0.03 (0.66) | 2.62+0.02 (0.77) | 2.05+0.02 (0.64) | 1.85+0.02 (0.6) |
| **Median** | 3.19+0.02 (0.86) | 2.63+0.01 (0.91) | 3.53+0.04 (0.63) | 3.02+0.03 (0.76) | 2.29+0.03 (0.57) | 2.06+0.02 (0.67) |
| **75th** | 3.90+0.02 (0.87) | 3.24+0.02 (0.89) | 4.77+0.06 (0.59) | 4.21+0.04 (0.71) | 2.73+0.04 (0.53) | 2.47+0.35 (0.57) |

The time series of the median OPC-N2 $PM_{2.5}$ concentrations along with the two reference
instruments are shown in Figure 5, and for a large portion of the inter-comparison all
instruments appear to be in reasonable agreement. However, there were a number of times
when the OPC-N2 readings were up to an order of magnitude higher relative to the reference
(e.g. 15th September), pointing to a significant instrument artefact. On the 15th September, the
GRIMM and TSI also move out of agreement and may point to the same artefact affecting the
GRIMM. Similar trends were also observed for the $PM_1$ and $PM_{10}$ mass fractions (Fig S4,
Supporting Information) with the OPC-N2 over-estimating the $PM_{10}$ concentration by several
orders of magnitude on 15th September (peak mass concentrations in the order of 15,000 µg
m$^{-3}$). Note that as EROS is an urban background site, it was unlikely to be affected by plumes





from sources such as vehicles and as a result these high concentrations spikes may not be
real.
The factors contributing to this apparent artefact shown by the OPC-N2 were investigated. In
Fig 6, the agreement between the OPC-N2 and the TSI instrument appears to vary as a
function of ambient RH, with reasonable agreement observed between the two instruments
during periods of relatively low ambient RH. However, during times when the RH was high
(>90%), the OPC-N2 recorded concentrations markedly higher than that measured by the TSI
3330 (Fig 6). Thus, it points to ambient RH as a significant contributing factor affecting the
particle mass concentrations measured by the OPC-N2, and this is tested further in later
sections. There are distinct differences in design in OPC-N2 compared to the reference
instruments (GRIMM and TSI 3330) as both the TSI 3330 and GRIMM utilise a sheath flow
unlike the OPC-N2. The sheath flow in both devices will be warmed to temperatures higher
than the ambient air due to proximity to the instrument pumps and electronics. This would
mean that they measure at a lower RH than ambient and could explain why no RH
dependence was observed on measured particle concentrations by the GRIMM and TSI 3330.

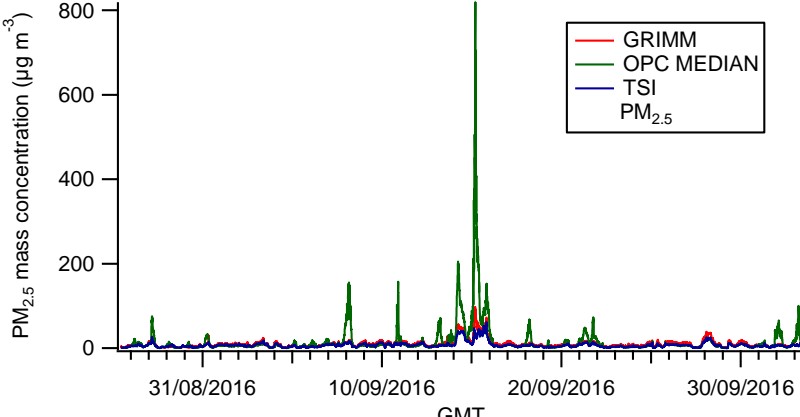

21   Figure 5: Time series of the measured $PM_{2.5}$ mass concentrations by the TSI, GRIMM and

22   median concentration measured by the14 OPC-N2 at EROS.





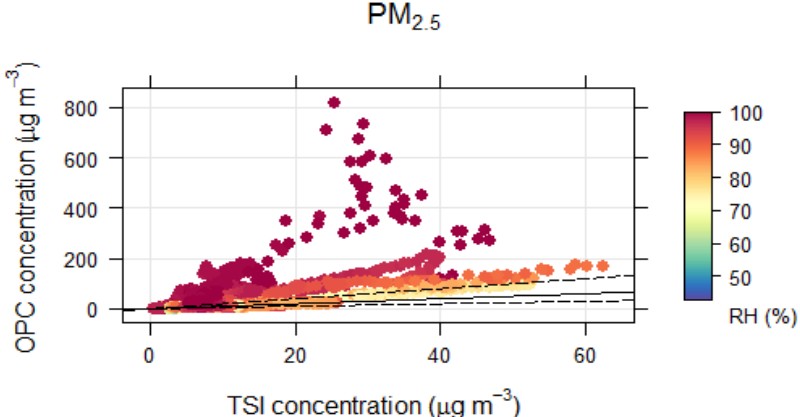

Figure 6: Measured concentrations by the TSI 3330 compared to the median concentration
measured by the 14 OPC-N2, coloured by the ambient relative humidity. Also shown are the
1:1 (solid) and 0.5:1 and 2:1 (dashed) lines.
**3.2.3 Comparison to TEOM-FDMS at AURN monitoring station**
We deployed a subset of the OPC-N2 devices (4) and the GRIMM at an urban background
AURN station, to enable comparison of the measured ambient particle mass concentrations to
a TEOM-FDMS. The time series of the measured concentrations of $PM_{10}$ and $PM_{2.5}$ for all
instruments is shown in Fig 7. The two reference instruments were found to be well
correlated ($r^2 > 0.91$, Figure S7, Supporting Information) but with the GRIMM reading was
about 20% lower than the TEOM, in agreement with previous work (Grover et al., 2006).
From Fig 6, periods of agreement between the four OPC-N2 and the reference instruments
(GRIMM and TEOM) were apparent, along with times when the four OPC-N2 measured
concentrations that were notably higher than the reference instruments. Overall, when
compared to the TEOM, the OPC-N2 measurements were 2.5-3.9 times higher for both the
$PM_{10}$ and $PM_{2.5}$, with considerable scatter observed (Table 2).



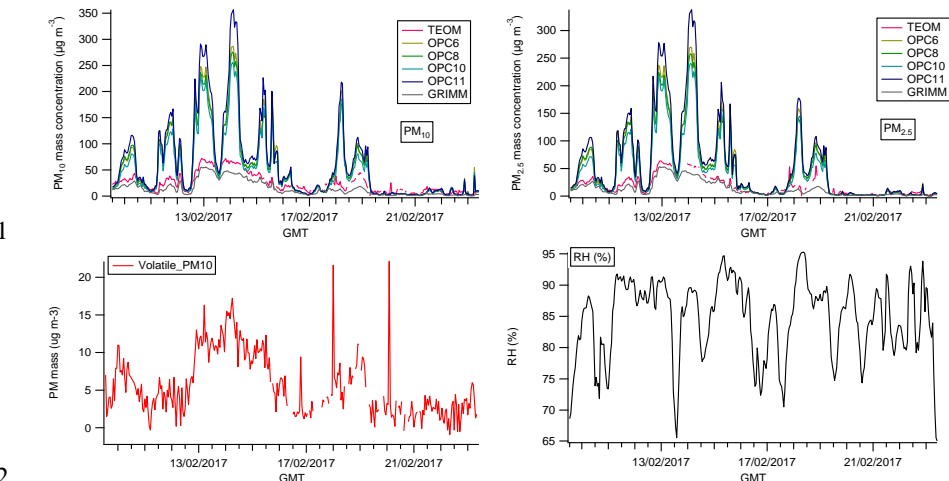

Figure 7: Time series for hourly measured PM mass concentrations by the TEOM, four OPC-
N2 and GRIMM at Tyburn Rd urban background AURN station. Relative humidity measured
at Tyburn Rd also shown.
Closer inspection of Fig 7 indicated that the times when the four OPC-N2 over-estimated the
particle mass concentrations were during times of high RH (e.g. 12-14[th] Feb), as observed in
the previous section. However, there were periods of high RH when the four OPC-N2 and
TEOM were in better agreement (e.g. 20[th] Feb onwards), indicating that the large positive
artefact observed in the OPC-N2 was not just related to RH. Rather, it appears that positive
artefact was observed during times when the volatile fraction measured by the TEOM was
relatively high, as well as higher RH, as was observed on 12-14[th] Feb (Fig 7). Thus, it
suggests that the ambient aerosol composition also contributed to the significant positive
artefact in the OPC-N2. A recent laboratory study found that the particle mass concentrations
measured by OPC-N2 for all three size fractions were highly linear with respect to
gravimetrically corrected reference instruments but that the slope was dependent on the
aerosol type (Sousan et al., 2016). Sousan et al. (2016) observed in the $PM_{10}$ fraction slopes
greater than unity for Arizona road dust but less than unity for salt and therefore suggest that
changes in aerosol composition may also account for the differences observed between the
reference instruments and OPC-N2 (Figs 7). This result highlights a limitation when
comparing optical methods to gravimetric - as differences may be due to changes in particle
mass, size distribution or composition: as all can affect the ability of a particle to scatter light
(Holstius et al., 2014).



From Fig 6, the times when there was a large positive artefact in the OPC-N2 occurred when
the RH was above 85%. If we exclude these times when the RH was over this threshold,
better agreement between the four OPC-N2 and the TEOM was observed, with slopes
between 1.1-1.7 for both size fractions (Table 2). One of the OPC-N2 recorded notably
higher mass concentrations compared to the reference instruments (OPC11), compared to the
other three OPC-N2 (Table 2), and this highlights the need to calibrate each OPC individually
before use in field measurements.
Table 2: Slopes of measured PM mass concentrations of the reference instruments (TEOM
and GRIMM) against the OPC-N2. The correlation co-efficient, $r^2$ is given in parenthesis.

|  |  | $PM_{10}$ | | | | $PM_{2.5}$ | | | |
|---|---|---|---|---|---|---|---|---|---|
|  |  | OPC6 | OPC8 | OPC10 | OPC11 | OPC6 | OPC8 | OPC10 | OPC11 |
| **ALL** | TEOM | 2.6 | 2.8 | 2.5 | 3.5 | 3.3 | 3.1 | 2.9 | 3.9 |
|  |  | (0.64) | (0.68) | (0.64) | (0.67) | (0.7) | (0.74) | (0.7) | (0.72) |
|  | GRIMM | 3.7 | 3.6 | 3.2 | 4.4 | 3.8 | 3.7 | 3.4 | 4.6 |
|  |  | (0.66) | (0.69) | (0.66) | (0.68) | (0.71) | (0.74) | (0.71) | (0.72) |
| **<85% RH** | TEOM | 1.4 | 1.4 | 1.2 | 1.7 | 1.3 | 1.4 | 1.1 | 1.6 |
|  |  | (0.82) | (0.83) | (0.83) | (0.83) | (0.79) | (0.8) | (0.79) | (0.79) |
|  | GRIMM | 1.8 | 1.9 | 1.6 | 2.2 | 2.0 | 2.1 | 1.7 | 2.4 |
|  |  | (0.83) | (0.84) | (0.84) | (0.84) | (0.89) | (0.89) | (0.9) | (0.88) |

## 3.3 Development of correction factor for ambient RH

Clearly there were times when there was a significant instrument artefact for the OPC-N2
(Figs 4 and S4) and the highest over-estimations occurred at high RH at both EROS and
Tyburn Rd (e.g. Fig 5 and 6). The size of hygroscopic particles is known to be dependent on
RH, as the particle refractive index and size are both a function of RH. Inorganic aerosols
(e.g. sodium chloride, nitrate and sulphate), make up a large portion of the $PM_{10}$ observed at
EROS (Yin et al., 2010), and are known to demonstrate an exponential increase in
hygroscopic growth at high RH (e.g. (Hu et al., 2010; Pope et al., 2010).





The ratio of measured mass concentrations by the OPC-N2 relative to the reference
instruments was plotted as a function of RH, and appeared to show an exponential increase
above ~85% RH, similar to hygroscopic particle growth curves (Pöschl, 2005). As a result,
we applied κ-Kohler theory (Petters and Kreidenweis, 2007), which describes the relationship
between particle hygroscopicity and volume by a single hygroscopicity parameter, κ. The κ-
Kohler theory can be adapted to relate particle mass to hygroscopicity at a given RH by
equation 5 (Pope, 2010):
$$a_w = \frac{(m/m_o - 1)}{(m/m_o - 1) + (\frac{\rho_w}{\rho_p}\kappa)}$$    (5)
Where $a_w$ is the water activity ($a_w$ = ambient RH/100), m and $m_o$ are the wet and dry (RH =
0%) aerosol mass, respectively. The density of the dry particles and water is given by $\rho_w$ and
$\rho_p$, respectively. The density of water is 1 g cm$^{-3}$, and the bulk dry particle density is assumed
to be 1.65 g cm$^{-3}$. The value for κ can be found by a non-linear curve fitting of a humidogram
(m/m$_o$ vs a$_w$), and was calculated using the TEOM measurements at Tyburn Rd in the first
instance as the TEOM system employs a Nafion dryer and so measures dry particle mass
(Grover et al., 2006). To account for the differences in mass concentration measured by the
TEOM and OPC-N2 at RH less than 85%, the scaling factors shown in Table 2 are used
calibrate the dry mass of the OPC-N2 to that observed in the TEOM, both in the PM$_{2.5}$ and
PM$_{10}$ fractions.
Figure 8 shows the humidogram plots, for both the PM$_{2.5}$ and PM$_{10}$ fractions, obtained by
plotting the ratio of OPC-N2 to the reference instrument (TEOM and GRIMM) outputs
versus RH.   When using the TEOM for m$_o$, similar κ constants were calculated for all OPC-
N2, ranging from 0.38-0.41 and 0.48-0.51 for PM$_{2.5}$ and PM$_{10}$, respectively, which is within
the expected range for Europe (0.36 ± 0.16, (Pringle et al., 2010). Similar κ values were
observed when using the GRIMM mass concentrations as the dry particle mass (m$_o$), ranging
from 0.41-0.44 and 0.38-0.41 for PM$_{2.5}$ and PM$_{10}$, respectively.






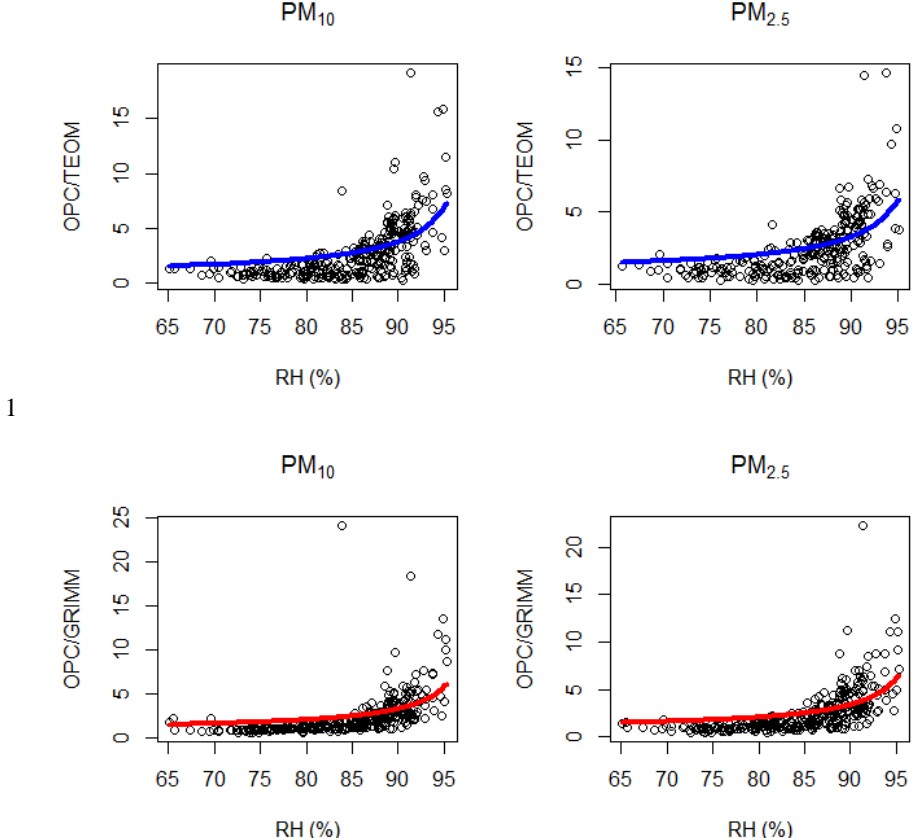

Figure 8: Measured and fitted humidograms ($m/m_o$ vs RH) recorded at the Tyburn Road
AURN site for $PM_{10}$ and $PM_{2.5}$ size fractions and reference instruments (TEOM and
GRIMM). The dry mass (m0) is given by the TEOM or GRIMM and the humidified mass is
given by the OPC-N2. Measured data is given by the black circles, the fitted data is given by
the blue (TEOM-FDMS) and red (GRIMM) line.
We then applied this fitting constant to model the expected OPC/Reference instrument ratio
for a given RH as a result of particle hygroscopic growth, by re-arranging Equation 5:
$$\frac{m}{m_o} = 1 + \frac{\frac{\rho_w}{\rho_p}\kappa}{-1+\frac{1}{a_w}} \qquad (6)$$



Where the $m/m_o$ is the ratio of the OPC-N2 to the reference instruments. Using Equation 6,
the mass concentrations measured by the OPC-N2 were corrected and significantly better
agreement between the corrected OPC-N2 and reference instruments was observed for
measurements across the whole range of ambient RH (Tables 2 and 3). Overall, the corrected
OPC-N2 mass concentrations using Eqn 6 were notably better, within 33% and 52% of the
TEOM and GRIMM, respectively. (Table 3) compared to 250-400% without the correction
factor (Table 2). The time series for the corrected data is shown in Figures S8 and S9
(Supporting Information) and there are periods were there is good agreement between TEOM
and the corrected OPC-N2.
There were also times when the OPC-N2 were clearly over-corrected (e.g. from 20[th] February
onwards), generally when the ambient RH was low (Fig 6). This suggests that when the RH
was below a threshold, Eqn 6 overcorrects the data and this can be observed in the
humidograms shown in Figure 8. Typically, at RH <80% the hygroscopic growth of real
atmospheric aerosols is small and it may be more appropriate to apply a linear regression
correction factor for data recorded under these RH conditions. During the period from the
20[th] February, the volatile particle fraction was also lower (Fig 6) and this indicates a
significantly different aerosol composition. Since $\kappa$ is composition dependent, a single global
fit to $\kappa$ will result in poor fitting when the true $\kappa$ is significantly different to the average $\kappa$.
The preceding discussion suggests that further refinement to the correction factors applied to
the OPC-N2 is possible, depending on the ambient RH and better knowledge of aerosol
composition. RH measurement is relatively trivial and can be achieved with small sensors but
aerosol composition determination still requires significant analytical equipment and
expertise.
Table 3: Summary of the comparison between the corrected OPC-N2 (via Eqn 6) against the
reference instruments

| OPC-N2 | TEOM | | GRIMM | |
|--------|-----------|-----------|-----------|-----------|
| | $PM_{2.5}$ | $PM_{10}$ | $PM_{2.5}$ | $PM_{10}$ |
| **OPC6** | 1.08±0.03 | 0.87±0.02 | 1.26±0.03 | 1.27±0.03 |
| **OPC8** | 1.11±0.03 | 0.89±0.02 | 1.29±0.03 | 1.23±0.03 |
| **OPC10** | 0.98±0.03 | 0.80±0.02 | 1.16±0.03 | 1.17±0.03 |
| **OPC11** | 1.33±0.04 | 1.06±0.03 | 1.53±0.04 | 1.51±0.04 |




1   **3.3.1 Longer-term monitoring with OPC-N2 at EROS**

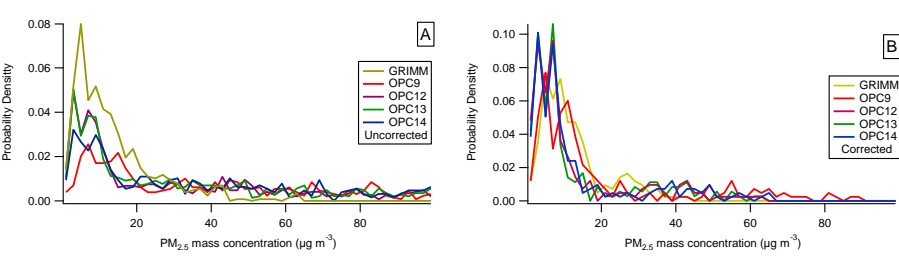

Figure 9: Histogram of measured PM$_{2.5}$ concentrations by the GRIMM PAS 1.108 and the 4
OPC-N2s for January. The uncorrected OPC-N2 concentrations are shown in the left plot
(A), while the right plot (B) shows the RH corrected OPC-N2 concentrations.
After the conclusion of the intensive measurements at EROS (Section 3.1), five of the OPC-
N2 continued monitoring for a further 4 months to examine if there was any evidence of
instrument drift over time, along with the GRIMM as reference. One of the OPC-N2 failed in
December, and so was excluded from this analysis. The remaining four OPC-N2 were
compared to GRIMM and in January after running for 4 months (Fig 9A), and while two of
the OPC-N2 had a similar distribution to the GRIMM (OPC13 and 14), the other two OPC-
N2 appeared to show evidence for instrument drift as the mode has shifted relative to the
GRIMM. However, the increased frequency of higher mass concentrations not observed by
the GRIMM but by all four OPC-N2 (Fig 9A) suggests that ambient RH is also a factor, as
the average RH in January (91%) was higher than September (84%). Therefore, we
calculated the correction for RH as described in the previous section (Eqn 6), as changes in
aerosol composition would affect the particle hygroscopicity. In addition, the κ was only
fitted for the data with RH < 95% since the hygroscopicity of aerosol is highly sensitive to
any error in the RH measurement above this value. Application of the RH correction factor
resulted in better agreement between each of the OPC-N2, with similar corrected
distributions observed (Fig 9B). Furthermore, the corrected OPC-N2 concentrations also had
better agreement with the GRIMM during January (Fig 9B) compared to uncorrected
concentrations (Fig 9A), suggesting that changes in the particle water content were the cause.





Thus, at least over a four month measurement period, there appears to be no evidence for
instrument drift in the OPC-N2, once appropriate correction factors were applied.
**3.4 Discussion on the OPC-N2 interferences**
In the previous sections, the significant positive artefact observed by the OPC-N2 relative to
the reference instruments were at times when the ambient RH was high, pointing to particle
water content as the cause. This result is perhaps not surprising, as many studies in the
literature have shown that particle water content can be a major reason for discrepancies
between techniques that measure ambient particle mass (See e.g. (Charron et al., 2004)). The
use of κ-Kohler theory to derive a correction factor based on ambient RH improved the
agreement between the OPC-N2 and reference instruments; however a limitation of this
approach is that the bulk aerosol hygroscopicity is related to particle composition, typically
the inorganic fraction (e.g. (Gysel et al., 2007)). Variation in ambient particle composition
could account for the large spread observed in the ratio of OPC-N2/TEOM at high RH (Fig
7), as an average hygroscopicity correction will overestimate when PM with higher
hygroscopicity is measured and vice versa. Furthermore, Eqn 6 may not be required for
locations where the ambient RH is lower than 85%, as typically atmospheric particle growth
due to water below this threshold is limited and a simple linear regression may be sufficient.
Thus, in-situ and seasonally specific calibrations for the OPC-N2 are required to account for
possible differences in ambient aerosol properties. However as κ values for continental
regions tend to fall within a narrow range globally (0.3±0.1, (Andreae and Rosenfeld, 2008),
with some systematic deviations for certain regions (Pringle et al., 2010), this average κ value
could be used in lieu of calibration with reference instrument (e.g. a TEOM) to determine the
correction factor (C) according to Eqn 7:
$C = 1 + \frac{0.3/1.65}{-1 + \frac{1}{a_w}}$     (7)

However, it should be noted that while *in situ* calibration of an OPC-N2 with suitable
reference instrumentation is preferable, for many locations around the world, and especially
low and middle income countries (LMICs), this may not be possible and so using an
appropriate κ value from the literature in Eqn 7 may be a reasonable approximation.



## 4.0 Applicability of OPC-N2 for ambient monitoring

The Alphasense OPC-N2 was evaluated for use in ambient monitoring of airborne particle mass concentration, with TEOM-FDMS and two commercial optical light scattering instruments; GRIMM PAS 1.108 and TSI 3330 employed as reference instruments. Comparison of the OPC-N2 to the reference optical instruments demonstrated reasonable agreement for the measured mass concentrations of $PM_1$, $PM_{2.5}$ and $PM_{10.}$ However, the OPC-N2 demonstrated a significant large positive artefact in measured particle mass during times of high ambient RH, and a calibration factor was developed based on bulk particle aerosol hygroscopicity. Application of the RH correction factor, based upon κ-Kohler theory, resulted in notable improvement with the corrected OPC-N2 measurements within 33% of a TEOM-FDMS. While higher than the slope of 1±0.1 allowed by US EPA, it is comparable to the agreement of a GRIMM to the TEOM (20%). All low cost PM sensors will likely require calibration factors to obtain the dry particle weight unless they actively dry the PM containing air stream before it enters the device. The use of heated inlets could be used to reduce the RH in the air stream but would have knock on consequences on the power requirements of the sensor, potentially making them less attractive for battery led operation. Reasonable precision between 14 OPC-N2 employed in the study was observed, with CV of 22+13% for $PM_{10}$ mass concentrations, with some of the variability likely due to use of separate but identical inlets.

Overall, while the OPC-N2 have been shown to accurately measure ambient airborne particle mass concentration provided they are correctly calibrated and corrected for RH. The reasonable level of precision demonstrated between multiple OPC-N2 suggests that they would be suitable for applications where a number of instruments are required such as spatial mapping and personal exposure studies.

## Acknowledgements

The authors wish to thank Peter Porter and Birmingham City Council for help in collocating the sensors next to the Tyburn Road AURN site. Funding is acknowledged from EPSRC (Global Challenges Research Fund IS2016). AL and MS acknowledge funding from the NERC National Capability programme ACREW and NE/N007115/1



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
