# Peer review of "Evaluation of a low-cost optical particle counter (Alphasense OPC-N2)"

_Atmospheric Measurement Techniques, 2017_

## Referee Comment (RC1) · Anonymous Referee #1 · 19 Sep 2017

This paper presents an evaluation of a low-cost sensor (OPC-N2) for monitoring ambient particulate matter. Three inter comparison field campaigns have allowed for determination of precision, comparison with reference instruments and suitability for long-term monitoring. This study gives new insights on the ability of these low-cost instruments to measure ambient particulate matter and notably, the identification and correction of bias related to high relative humidity conditions. The manuscript is clear, well-written and is suitable for publication after considering minor changes.

Figure 3 shows that calculated average coefficients of variance (CV) (line 17 page 9) are influenced by a few high values and are below 0.1 most of the time. This effect of a few high values on average CV should be considered in the discussion. Would it be possible to quantify the bias due to relative humidity?

[Figure]

Determination of K value: please detail the calculation of K and its uncertainties. Humidograms on Figure 8 show that fitted models may possibly be not suitable. When fitted curves are used for prediction or for quantification, quality and suitability of fitted models need to be examined through an analysis of residues. Here I expect that the model is not suitable (overestimation at low RH and conversely). In this case this would support the assumption page 19 of the necessity to use two models for low and high relative humidities - that would improve the correction independently of aerosol composition.

Many figures are small and difficult to read and assess (1 ; 5 ; 6 ; 7 ; 9 ; S2 ; S4 ; S5). In particular for readability Figure 1 could be reduced to the second period (21st to 24th of September) and Figures 5 and S5 need to be re-scaled since most concentrations are flattened by a few very high values.

Tables 1, 2 and 3: are they results of linear regressions (slopes) or ratios? In the first case indicate if intercepts are strained to zero or are non-significant.

The statement lines 16-17 page 20 (while two of the OPC-N2 had a similar distribution to the GRIMM (OPC13 and 14), the other two OPC-N2 appeared to show evidence for instrument drift as the mode has shifted relative to the GRIMM) is not obvious from Figure 9.
* * *

---

## Referee Comment (RC2) · Anonymous Referee #2 · 4 Oct 2017

This manuscript describes the evaluation of a low cost optical particle sensor with respect to ambient PM monitoring. The advent of such low cost sensors is an important development in the PM monitoring field which will be important for future spatial distribution measurements and hence epidemiological health studies. The topic is well within the scope of AMT, and could be useful to community in understanding the advantages and limitations of such technology. However, the manuscript is not entirely well written, suffering at times from lack of clarity, and incomplete information. The issues are described further below. If these issues can be addressed then I believe this manuscript could be publishable in AMT and provide useful information.

Overall, the manuscript is too qualitative with respect to understanding how accurate and precise these sensors may be. On too many occasions the authors use the terminology "reasonable" to describe the agreement or precision etc.. Such terminology is far too subjective. What is considered "reasonable"? The authors should strive to be more quantitative in this respect, as many people will want to use such sensors and their recommendation may carry some weight within the community.

In my opinion, such technology has a long way to go before it can be a useful in determining the spatial distribution of PM and hence be used in health studies. One could argue that the accuracy is less important than the inter-instrument variability in this regard. However, a CV between sensors varying from 0.2 to 0.8 does not inspire confidence (ie. fig 3). The authors seem to think that such a CV is adequate, however if that is the case they must justify why they think that to be "reasonable". On pg 9, line 20 the author's state that the CV is "perhaps not unreasonable". This is entirely speculative, and depends upon the application. For most applications I doubt this is reasonable. The authors overall seem to be saying that this is a good sensor for deployment for spatial/health studies, when in reality the data they show indicate that is not really the case. I suggest this technology remains quite far from easily being used in such studies, especially because of the variability between instruments, the need for corrections on individual instruments, and the poor accuracy. These limitations need to be front and center in this manuscript to avoid confusion.

The comparison of the sensor with a TEOM needs to be justified more concretely. It is not clear how they can be comparing "apples-to-apples" with a TEOM which by their own admission uses a nafion dryer to dry particles first (while the OPC does not). The authors should explain exactly what the TEOM they are using is providing and how it can be compared to the OPC sensor. Are they truly comparing the same quantity? At first glance it does not seem like they are, but not enough information is provided to determine this. For that matter, why are they comparing with a TEOM at all, if they have just finished assessing the accuracy with a TSI/GRIMM. By doing so, they are adding another uncertain variable into the assessment which may not be needed.

The description of the OPC sensor that is being investigated is highly lacking information. The authors need to improve their description of the sensor significantly. Although it may have been described in other work (which they have not even cited), it should be in part described here as well. Reading this short paragraph description I am left wondering: How does it sample? With a pump? Passively? How does the data collection work? What data is collected exactly? Does it only provide a mass concentration value? Does it provide number concentrations as well? What is the time resolution? What does the manufacturer say it should do? All these things and likely more need to be described in the methods section.

If the GRIMM instrument is noted to always be 20% higher than the TSI, then which one is the standard? I am assuming that the TSI is the so-called "gold standard", as it is calibrated with a known stream of particles at some point or another. Is that the case? The authors make it sound as if they realize that the GRIMM is consistently incorrect. If so, then why are they using the GRIMM as a comparison at all? If they are trying to assess the accuracy of the OPC then they should determine which standard is truly accurate, and only compare to one of them. It does not make sense to me to be assessing accuracy with an instrument which is not providing the correct values. It seems the true measure of accuracy is using the TSI, so why not simply use that?

Since the reference instruments and the OPC are essentially coarse particle instruments, the inlet fabrication and geometry are critical in transmitting the largest particles into any of these instruments. Any slight bends and differing bends between instruments will highly impact the large particles that enter the instruments. How is this mitigated? Are they the same between standards and the OPSs? If not, then I don't see how any real analysis of accuracy can be made, since some large particles being lost preferentially can severely affect the PM10 mass. The authors could potentially calculate the losses as a function of size and inlet bends etc. . .using on-line calculators at the very least, to be sure they are at least consistent between instruments. This is less of a concern for the precision determination.

While I do not doubt that the OPC has an artefact associated with RH, I also notice

in many of the figures that the inaccuracy seems to be worse at higher PM loading. Is it possible that the high RH may also be correlating with high mass? In that case which one is more important? Is it truly the RH or is it the mass that is causing the artefact? By their own admission, the authors note that there are other factors at play. Can these factors be determined? It would seem that rather than a correction based only on Kholer theory, additional corrections are needed. It might be possible to make a multivariate empirical correlation between the OPC/TSI ratio and the RH, mass, and/or others. Can this be done? A multivariate analysis may help to determine what factors are truly responsible for the discrepancy and to what degree.

It remains unclear why RH should cause an artefact. I do not dispute that one exists, but the authors should attempt to explain why fundamentally the RH should make any difference to the OPC. In principle the OPC is determining if a particle scatters or not. If it does, then it is counted. So even if RH affects scattering (which it will), then I do not see how it will stop the scattering all together such that a particle is not counted. The authors need to provide a plausible hypothesis at least to explain this issue.

What does the manufacturer say the specifications should be for the OPC sensor? It seems like no attempt was made to contact the manufacturer to get an idea of how the mass is calculated. Given they are assessing their instrument; one would think they would be agreeable to helping them out. How do these results compare with what the manufacturer says it should do in terms of accuracy and precision?

There are many studies where mobile measurements of PM were made in urban and suburban areas. By looking at the spatial variation of the PM in those studies, one can get an idea of what kind of inter-instrument variability is required for this to be a useful instrument. Some attempt at this should be done, at least qualitatively.

Minor issues:

Pg 2, line 2: the term "reasonable" is used here and not justified.

[Figure]

Pg 2, line 30: this line is awkwardly written. Remove the "are" and use "companies" or "manufacturers" but not both.

Pg 3 , line 20: define "PUWP" and "dylos"

Pg 3, line 19: add "the" before "dylos" (if I am reading this correctly)

Pg 3, line 21: remove the "s" from "environments"

Pg 3, line 22: add "they" after "however"

Pg 3, line 29: "sites" to "site"

Pg 4, line 11: replace "were" with "used"

Pg 4, line 15-17: awkwardly written. Please improve. And remove "s" from "systems"

Pg 5, line 17-18: it is not clear what this is supposed to be used for in this paper.

Pg 5, line 29: briefly describe what the point of the "filter dynamic system" is.

Pg 6, line 6: add an "s" to "OPC"

Pg 8, line 15: awkwardly written. Please improve.

Pg 9, line 20: far too speculative without backing it up.

Pg 10, line 5: define what "consistent" means to you. Fig 3 indicates it is not at all consistent. . .

Pg 10, line 7: again, "reasonable" is too subjective.

Pg 10, line 23: again, the use of "reasonable". . ..what does this mean?

Pg 11, line 5: it should not agree with the GRIMM as you have already stated it is 20% off to begin with.

Pg 15: how is the volatile fraction determined? (briefly). What does "gravimetrically corrected" mean in this context?

Table 2: units of slope? Or unitless?

Pg 17, line 1: is this the median of all OPCs or all them individually?

Pg 21, lines 7-8: this has no bearing on the current study.

Pg 22, line 15: what is "knock on"??

Pg 22, line 20: remove "while"

Pg 22, line 23: "suitable" is not what the reader gets from this paper. See my comments above.

Figure 1: difficult to see as there are too many lines. Perhaps shorten the time scale and zoom in. Perhaps a log scale would help too.

Figure 5: too small to see anything other than the peak. Perhaps use a log scale to better see what is going on.

Figure 6: Too small to see anything. I suggest you split the y-axis and zoom in to where the majority of data is.

————————————————————

---

## Short Comment (SC1) · 23 Oct 2017

Albeit briefly, European Standard EN481 is mentioned in the OPC-N2 user manual when describing how PM is calculated from the particle number concentration data.

The author could be more specific about the inlet arrangements with their use of the OPC-N2. In addition to comments made in this subject by referee RC2, with its small fan aspirator, the air-flow through the device may easily affected by changes to its default inlet or the nature of the ambient air e.g. breeze across the inlet. Possible differences in response between these and the reference instruments due to such factors should be discussed.

---

## Author Comment (AC1) · 22 Nov 2017

We thank the anonymous reviewers and W.R. Stanley for their time and insightful comments. We believe their help has made the paper significantly better. In the attached file we respond to the all reviewer's comments one by one. We also show the updated paper and supplementary information, which have been highlighted where changes were made.

Please also note the supplement to this comment:
https://www.atmos-meas-tech-discuss.net/amt-2017-308/amt-2017-308-AC1-supplement.pdf

---

## Author Comment (AC2) · 22 Nov 2017

**Received and published: 19 September 2017**

This paper presents an evaluation of a low-cost sensor (OPC-N2) for monitoring ambient particulate matter. Three inter comparison field campaigns have allowed for determination of precision, comparison with reference instruments and suitability for long-term monitoring. This study gives new insights on the ability of these low-cost instruments to measure ambient particulate matter and notably, the identification and correction of bias related to high relative humidity conditions. The manuscript is clear, well-written and is suitable for publication after considering minor changes.

1. Figure 3 shows that calculated average coefficients of variance (CV) (line 17 page 9) are influenced by a few high values and are below 0.1 most of the time. This effect of a few high values on average CV should be considered in the discussion. Would it be possible to quantify the bias due to relative humidity?

**Response:**

The mean CV for the times when the RH was less than 85%, when we typically observed little influence from ambient RH on the measured particle mass concentration by the OPC-N2, was  $0.3\pm0.25$ ,  $0.23\pm0.14$  and  $0.2\pm0.18$  for PM1, PM2.5 and PM10 mass concentrations, respectively. This was only slightly lower than the overall average (at all RH:  $0.32\pm0.16$ ,  $0.25\pm0.14$  and  $0.22\pm0.13$  for PM1, PM2.5 and PM10 mass concentrations, respectively). However, while we observed higher CV when the RH was above 85% ( $0.34\pm0.30$ ,  $0.27\pm0.14$  and  $0.23\pm0.21$  for PM1, PM2.5 and PM10 mass concentrations, respectively). However, while we observed higher CV when the RH was above 85% ( $0.34\pm0.30$ ,  $0.27\pm0.14$  and  $0.23\pm0.21$  for PM1, PM2.5 and PM10 mass concentrations, respectively) and suggests that the individual OPC-N2 responded slightly differently to the effect of RH these were within the variability. Thus it suggests that ambient RH did not affect the precision of the OPC-N2 significantly.

We now state in the paper that the following on p17 L20 "Whilst the accuracy of the instrument was significantly worse at high RH the precision remains the same within error. The CV analysis conducted in section 3.1.2 is repeated for the same dataset but put into low (RH<85%) and high RH (RH>85%) subsets. For high RH conditions the CV for PM1, PM2.5 and PM10, was 0.34±0.30, 0.27±0.14 and 0.23±0.21, respectively. For low RH conditions the CV for PM1, PM2.5 and PM10, was 0.30±0.25, 0.23±0.14 and 0.20±0.18, respectively."

2. Determination of K value: please detail the calculation of K and its uncertainties. Humidograms on Figure 8 show that fitted models may possibly be not suitable. When fitted curves are used for prediction or for quantification, quality and suitability of fitted models need to be examined through an analysis of residues. Here I expect that the model is not suitable (overestimation at low RH and conversely). In this case this would support the assumption page 19 of the necessity to use two models for low and high relative humidities - that would improve the correction independently of aerosol composition.

**Response:**

We have repeated the analysis using two models as suggested by the reviewer, a linear correction for times when the ambient low RH was low (<85%) and for times at higher RH (>85%) a fitting based upon  $\kappa$ -Kohler theory (Eqn 6). We then compared the results of using this binary two model approach, to that originally applied, using Eqn 6 for all ambient RH. The results are shown below as scatterplots of the corrected OPC-N2 against the TEOM concentrations for PM2.5. As can be seen Figure 1, there was little improvement in the slope or  $r^2$  with the two model correction (Cv2) compared to the using correction with Eqn 6 for all RH (C). What was noticeable was that the intercept for the two model approach (Cv2) moved closer to zero, suggesting that at the lower mass concentrations the correction was improved. Similar trends were also observed for PM10.

Figure 1: Scatterplots of corrected OPC-N2 against the TEOM for PM2.5 mass concentrations. The two model approach (Cv2) is in red and the one model approach in blue.

Therefore we have added the following text at page 20, line 15:

"There were also times when the OPC-N2 were clearly over-corrected (e.g. from 20th 10 February onwards), generally when the ambient RH was low (Fig 6). This suggests that when the RH was below a threshold, Eqn 6 overcorrects the data and this can be observed in the humidograms shown in Figure 8. Typically, at RH <85% the hygroscopic growth of real atmospheric aerosols is small and it may be more appropriate to apply a linear regression correction factor for data recorded under these RH conditions. Therefore we applied a binary two model approach to correct the OPC-N2 mass concentrations, where a linear correction (using the TEOM as reference concentration) for when RH <85%, and above this threshold in RH Eqn 6 was used. As can be seen Figure S9 (Supporting Information), there was little change in the slope or  $r^2$  value with the two model correction compared to the using correction with Eqn 6 for all RH. What was noticeable was that the intercept for the two model approach moved closer to zero, suggesting that at the lower mass concentrations the correction was improved. Similar trends were also observed for PM10."

3. Many figures are small and difficult to read and assess (1;5;6;7;9;S2;S4;S5). In particular for readability Figure 1 could be reduced to the second period (21st to 24th of September) and Figures 5 and S5 need to be re-scaled since most concentrations are flattened by a few very high values.

Response

We have fixed figures 1, 5, 7, 9, S2 and S4 as suggested. We have not rescaled Fig 6 and S5 as the point of this figure is to show that there are times when the OPC-N2 over-estimated the PM mass concentration over a very large scale, and rescaling the y-axis would lose this information.

4. Tables 1, 2 and 3: are they results of linear regressions (slopes) or ratios? In the first case indicate if intercepts are strained to zero or are non-significant.

Response:

All presented relationships are linear regressions (slopes). The intercepts were not constrained to zero and they vary from instrument to instrument. In Table 1, similar intercepts were observed for relationships between the OPC-N2 and TSI and GRIMM, and were around -1, -12 and -10 for PM1, PM2.5 and PM10, respectively. For Table 2, the intercepts against the TEOM were also significant, at -12 and -15 for PM2.5 and PM10, respectively. The intercepts were found to be notably improved with the application of the RH correction (Table 3), and were around zero for the GRIMM (both size fractions) and about -3 for the TEOM.

We chose not to constrain the regression to zero to not bias the analysis, and the significant negative intercepts likely reflect the influence of a few high measurements by the OPC-N2 in Tables 1 and 2.

We have included this information in the headings for Tables 1-3, with the new heading for Table 1 shown as an example:

"Table 1: Slopes (linear regression) of measured PM mass concentrations of the reference instruments against the median and inter-quartiles for OPC-N2. The intercepts were not constrained to zero. Correlation co-efficient, r2 is given in parenthesis.

5. The statement lines 16-17 page 20 (while two of the OPC-N2 had a similar distribution to the GRIMM (OPC13 and 14), the other two OPC-N2 appeared to show evidence for instrument drift as the mode has shifted relative to the GRIMM) is not obvious from Figure 9.

Response:

We have adjusted this sentence to now read:

"The remaining four OPC-N2 were compared to GRIMM and in January after running for 4 months (Fig 8A), and while three of the OPC-N2 had a similar distribution to the GRIMM (OPC12, 13 and 14), OPC9 appeared to show evidence for instrument drift as the mode has shifted relative to the GRIMM."

We have also added the following sentence to the conclusions to highlight this apparent instrument drift

"One out of four OPC-N2 tested for long-term monitoring appeared to show evidence for instrument drift relative to reference instruments."

**Anonymous Referee #2**

Received and published: 4 October 2017

This manuscript describes the evaluation of a low cost optical particle sensor with respect to ambient PM monitoring. The advent of such low cost sensors is an important development in the PM monitoring field which will be important for future spatial distribution measurements and hence epidemiological health studies. The topic is well within the scope of AMT, and could be useful to community in understanding the advantages and limitations of such technology. However, the manuscript is not entirely well written, suffering at times from lack of clarity, and incomplete information. The issues are described further below. If these issues can be addressed then I believe this manuscript could be publishable in AMT and provide useful information.

1. Overall, the manuscript is too qualitative with respect to understanding how accurate and precise these sensors may be. On too many occasions the authors use the terminology "reasonable" to describe the agreement or precision etc.. Such terminology is far too subjective. What is considered "reasonable"? The authors should strive to be more quantitative in this respect, as many people will want to use such sensors and their recommendation may carry some weight within the community.

**Response:**

The term reasonable was used to make the paper more readable. We did throughout the paper apply quantitative analysis of the accuracy and precision of the OPC-N2 such as CV (Fig 3) and comparison to reference instrument (slopes by linear regression, e.g. Tables 1-3) and therefore feel we have provided this information. Low-cost sensors are by their nature a compromise between cost and quality and therefore we don't believe that the same criteria for research or monitoring grade instruments should necessarily apply when considering their performance.

The reviewer does make a valid point that we should define what we mean by reasonable and as a result we have made a number of changes the text in the following locations to address this:

Abstract, page 1 line 30:

"Inter-unit precision for the 14 OPC-N2 sensors of 22±13% for PM10 mass concentrations was observed"

page 3, line 10:

"Laboratory assessments of the performance of a number of low-cost miniature OPC's have shown promising results, with adequate precision observed when compared to reference instrumentation (Manikonda et al., 2016)."

**Page 2, line 2:**

"The level of precision demonstrated between multiple OPC-N2 suggests that they could be suitable device for applications where the spatial variability in particle concentration was to be determined, but need characterisation."

Page 6 line 24:

"which is not strictly true for airborne particles in an urban atmosphere but is considered a standard approximation."

Page 13, line 9:

"The time series of the median OPC-N2  $PM_{2.5}$  concentrations along with the two reference instruments are shown in Figure 5, and for a large portion of the inter-comparison all instruments appear to be in agreement."

*Page 13, line 22:*

"In Fig 6, the agreement between the OPC-N2 and the TSI instrument appears to vary as a function of ambient RH, with better agreement observed between the two instruments during periods of relatively low ambient RH."

Page 23, line 22

"Comparison of the OPC-N2 to the reference optical instruments demonstrated reasonable agreement for the measured mass concentrations of  $PM_1$ ,  $PM_{2.5}$  and  $PM_{10}$  as evidenced by the stated accuracy and precision."

2. In my opinion, such technology has a long way to go before it can be a useful in determining the spatial distribution of PM and hence be used in health studies. One could argue that the accuracy is less important than the inter-instrument variability in this regard. However, a CV between sensors varying from 0.2 to 0.8 does not inspire confidence (ie. fig 3). The authors seem to think that such a CV is adequate, however if that is the case they must justify why they think that to be "reasonable". On pg 9, line 20 the author's state that the CV is "perhaps not unreasonable". This is entirely speculative, and depends upon the application. For most applications I doubt this is reasonable. The authors overall seem to be saying that this is a good sensor for deployment for spatial/health studies, when in reality the data they show indicate that is not really the case. I suggest this technology remains quite far from easily being used in such studies, especially because of the variability between instruments, the need for corrections on individual instruments, and the poor accuracy. These limitations need to be front and center in this manuscript to avoid confusion.

**Response:**

We note that Reviewer 1 agreed with our interpretation of Fig 3.

We don't agree that the OPC-N2 is far away from being useful in mapping spatial distribution of PM, but rather that the results presented in this paper show that this sensor can provide an accurate measure of PM concentration provided they are calibrated against reference instrument and the corrected for the RH artefact. All instruments need calibration if they are to provide useful data, and the OPC-N2 is no exception and we believe that the correction factors presented in this work enable the OPC-N2 to provide more accurate measurements. This is most explicitly evidenced in Figs S8-9, where we observed notable improvement in agreement, not just with reference instrument (in this case the TEOM) from 250-400% to 33%, but also between the four OPC-N2. We believe the proposed correction factor using k-theory is a significant advance in the use of these low cost OPCs, as we stated in Section 3.4, page 22, line 28:

"The use of  $\kappa$ -Kohler theory to derive a correction factor based on ambient RH improved the agreement between the OPC-N2 and reference instruments; however a limitation of this approach is that the bulk aerosol hygroscopicity is related to particle composition, typically the inorganic fraction (e.g. (Gysel et al., 2007)). Variation in ambient particle composition could account for the large spread observed in the ratio of OPC-N2/TEOM at high RH (Fig 7)"

The precision of these instruments was stated within the text (Section 3.1.2) where on average, the 14 OPC-N2 were found to have CV of 22±13% for PM10 mass concentrations without any corrections applied, with only the occasional spike in CV evident in Fig 3. In Fig 3, these are uncorrected results and the cause of the spikes in CV appears to be a result of the aforementioned RH

artefact (Please see our response to Reviewer 1, comment #1), as each OPC-N2 was found in Section 3.2 and 3.3 to respond differently to RH artefact.

On page 10, line 11 we used the phrase that this CV between the 14 OPC-N2 was "perhaps not unreasonable for low-cost sensor" as low-cost sensors are by nature a compromise between cost and quality. Much of the literature to date has focused on the accuracy of low-cost sensors and consequently there is little literature quantifying their precision. Sousan et al. (2016) reported for their laboratory measurements, the OPC\_N2 had a CV of between 4.2-16%, which while lower than the current work, this would be expected for a controlled environment.

Recent work by Lewis et al., (2016) showed that 20 unspecified PM sensors sampling ambient air had an inter-quartile range of around 20 ug m-3, significantly higher than the current work. Wang et al. (2015) reported for three low-cost light scattering particle sensors, standard deviations of 15-90  $\mu$ g m-3 for repeated laboratory measurements of concentrations up 1000  $\mu$ g m-3. With limited comparable studies to assess the precision of the OPC-N2, we can only present our findings but suggest that the precision of the OPC-N2 is significantly improved compared to Lewis et al. for ambient measurements.

*In addition, the precision and accuracy of the OPC-N2 was also found to improve with the application Eqn 6, as shown by Table 3, Figures 8, S8-9, compared to uncorrected concentrations.*

The reviewer does make a valid point that we should avoid subjective terms (such as reasonable) and as such we have made several changes to the text to avoid this, please see our response to the previous comment.

Furthermore, we did state within in the text (page 10, line 13) "precision of the OPC-N2 would need to be considered when comparing multiple units", so we did highlight this issue in the text. However, we agree that we should make this point clear and so we have altered Section 4.0 (page 24, line 2) to highlight this issue:

"For  $PM_{10}$  mass concentrations, a CV of 22+13% between the 14 OPC-N2 employed in this study was observed, with some of the variability likely due to use of separate but identical inlets, and therefore could be considered reasonable for a low-cost sensor but this level of precision needs to be considered when using multiple units."

3. The comparison of the sensor with a TEOM needs to be justified more concretely. It is not clear how they can be comparing "apples-to-apples" with a TEOM which by their own admission uses a nafion dryer to dry particles first (while the OPC does not). The authors should explain exactly what the TEOM they are using is providing and how it an be compared to the OPC sensor. Are they truly comparing the same quantity? At first glance it does not seem like they are, but not enough information is provided to determine this. For that matter, why are they comparing with a TEOM at all, if they have just finished assessing the accuracy with a TSI/GRIMM. By doing so, they are adding another uncertain variable into the assessment which may not be needed.

**Response:**

We compared against a TEOM instrument as this an accepted regulatory standard instrument for particle mass measurements, and in particular was being run as part of the UK monitoring network . As we stated in the text (page 5 line 27), this is not a 'apples-to-apples' comparison, we were aware that the OPC-N2 and TEOM are fundamentally two different techniques and so there would be disagreement. But they are two approaches to the same measurement and we were interested to see how the calculated particle mass concentration by the OPC-N2 compared to a more direct measurement of particle mass concentrations by the TEOM. We note that many previous studies have compared optical particle counters to TEOM to see how the particle mass

measurements from optical particle counter instruments compare with the assumptions made to convert particle number concentrations to mass (see e.g. Wang et al 2016).

*Therefore, we feel we are justified in comparing the measured particle mass by the OPC-N2 to a TEOM.*

4. The description of the OPC sensor that is being investigated is highly lacking information. The authors need to improve their description of the sensor significantly. Although it may have been described in other work (which they have not even cited), it should be in part described here as well. Reading this short paragraph description I am left wondering: How does it sample? With a pump? Passively? How does the data collection work? What data is collected exactly? Does it only provide a mass concentration value? Does it provide number concentrations as well? What is the time resolution? What does the manufacturer say it should do? All these things and likely more need to be described in the methods section.

**Response;**

The OPC-N2 samples via a small fan, and can sample at min time resolution of 10s. The OPC-N2 is described in more detail in Sousan et al. (2016), and we have added reference to this paper. As we described in the second paragraph of Section 2.1.1, the OPC-N2 has been designed to log via Alphasense software on a laptop, and is also where we describes the custom built logging system we built.

The OPC-N2 collects number concentration and converts this to mass concentration via on-board factory calibration, as we describe in detail in Section 2.3.

Number concentration per size bin is available by size bin but we chose to focus on the mass concentration as this is the output that majority of users of an OPC-N2 are likely to use.

To include this additional information, the first paragraph of Section 2.1.1 (page 4, line 8) now reads:

"The Optical Particle Sensor (OPC) under evaluation in the current work is the OPC-N2 manufactured commercially by Alphasense (www.alphasense.com) and is described in detail in Sousan et al. (2016). The OPC-N2 can be considered as a miniaturized OPC as it measures 75x60x65 mm and weighs under 105 g, and as such is significantly cheaper (approx. £200) than the comparable reference instruments (see next section). The OPC-N2 samples via small fan aspirator and measures particle number concentration over a reported size range of 0.38 to 17 µm across 16 size bins, and maximum particle count of 10,000 per second. The minimum time resolution is 10s. The measured particle number concentration is converted via on-board factory calibration to particle mass concentrations for PM1, PM2.5 and PM10 size fraction according to European Standard EN481 (OPC-N2 manual). According the OPC-N2 manual, the standard definition for PM10 in EN 481 extends beyond the particle size measured by the OPC-N2, and may consequently underestimate PM10 value by up to 10%. Further discussion on calculations for conversion from particle number to mass concentrations is given in Section 2.3. All OPC-N2 in this study were firmware version 18."

5. If the GRIMM instrument is noted to always be 20% higher than the TSI, then which one is the standard? I am assuming that the TSI is the so-called "gold standard", as it is calibrated with a known stream of particles at some point or another. Is that the case? The authors make it sound as if they realize that the GRIMM is consistently incorrect. If so, then why are they using the GRIMM as a comparison at all? If they are trying to assess the accuracy of the OPC then they should determine which standard is truly accurate, and only compare to one of them. It does not make sense to me to be assessing accuracy with an instrument which is not providing the correct values. It seems the true measure of accuracy is using the TSI, so why not simply use that?

**Response:**

Both the GRIMM and TSI 3330 are accepted and widely used instruments for measuring particle number size distribution, and we are not claiming that the TSI is the gold standard. The GRIMM is an instrument that is has been designated a federal equivalent method (FEM) for measuring particle mass concentrations by the US EPA, and as such we do not consider it inferior to the TSI 3330.

That they did not agree is not entirely unexpected, as while there are usually excellent correlation, the slopes are rarely unity between different optical particle counters (See e.g. (Castellini et al., 2014; Dinoi et al., 2017).

Therefore as both TSI and GRIMM are widely used and airborne particle measurements are inherently instrument dependent, we chose to compare to both instruments in this study to see if there were any differences.

6. Since the reference instruments and the OPC are essentially coarse particle instruments, the inlet fabrication and geometry are critical in transmitting the largest particles into any of these instruments. Any slight bends and differing bends between instruments will highly impact the large particles that enter the instruments. How is this mitigated? Are they the same between standards and the OPSs? If not, then I don't see how any real analysis of accuracy can be made, since some large particles being lost preferentially can severely affect the PM10 mass. The authors could potentially calculate the losses as a function of size and inlet bends etc, using online calculators at the very least, to be sure they are at least consistent between instruments. This is less of a concern for the precision determination.

**Response:**

At EROS for the intensive inter-comparison all 14 OPC-N2 were fitted with a 12cm long stainless steel tubing that sampled horizontally at the same height (1.5m). The TSI 3330 and GRIMM also sampled at the same height. The GRIMM has a horizontal inlet that connects to black conductive tubing, which was of a similar length. The TSI has meanwhile has a vertical inlet and due to inlet constraints in a bend in the conductive tubing was necessary. Due to size of the inlets on the instruments, they were different diameters, 3/8" for OPC-N2 and ¼" for TSI and GRIMM. As a result of the above, we could not use the same length tubing or orientation for each instrument and while aware of this potential for different particle sampling efficiencies but were restricted by practicalities of the sampling location.

As suggested by the reviewer, we calculated the expected particle loss in a sample lines (using an on-line calculator, (Von der Weiden et al., 2009) for the TSI as it was the only one with bend in the inlet. With the sampling set up we used, we calculated a sampling efficiency of 92% for 10  $\mu$ m particles.

We have added this additional information to the text at Section 2.2.1 (page 5, line 23):

"Minimal lengths (12cm) of stainless steel tubing (OPC-N2) and conductive black tubing (TSI 3330 and GRIMM) were used to sample outside air, with each OPC having its own inlet at a height of 1.5 m. The vertical inlet for the TSI 3330 necessitated a bend in the tubing, however the calculated sampling efficiency (using von der Weiden et al., 2009) was 92% for particles with a diameter of 10  $\mu$ m. Therefore, while the inlet arrangement of the TSI 3330 may have affected the intercomparison, particularly when considering the accuracy of the OPC-N2, we were limited to what was practical."

The TEOM by design has a vertical inlet, and so we placed the OPC-N2 for this comparison as close to the TEOM inlet as possible on the roof, using the same length inlet as the intensive inter-comparison in September and so we believe should not overly affected the inter-comparison.

6. While I do not doubt that the OPC has an artefact associated with RH, I also notice in many of the figures that the inaccuracy seems to be worse at higher PM loading. Is it possible that the high RH may also be correlating with high mass? In that case which one is more important? Is it truly the RH or is it the mass that is causing the artefact? By their own admission, the authors note that there are other factors at play. Can these factors be determined? It would seem that rather than a correction based only on Kholer theory, additional corrections are needed. It might be possible to make a multivariate empirical correlation between the OPC/TSI ratio and the RH, mass, and/or others. Can this be done? A multivariate analysis may help to determine what factors are truly responsible for the discrepancy and to what degree.

**Response:**

We think that it is the RH that is causing the artefact not the particle mass and we feel that this was best evidenced by Figure 6. For a given range of RH, we did not observe a curve as would be expected if there was mass loading effect, rather a straight line. This strongly suggests that RH was the cause. The artefact at high RH was due to particle hygroscopicity, and so will also be affected by the particle composition. This was likely why there were times at high RH when the OPC are in better agreement with the reference instruments (See e.g. Fig 7).

While it is likely possible to make a correction factor based upon the RH and particle composition, as we discussed in Section 3.4 for this study we did not have access to on-line measurements of particle composition, so we cannot formulate this correction factor. This will be the focus of future work.

7. It remains unclear why RH should cause an artefact. I do not dispute that one exists, but the authors should attempt to explain why fundamentally the RH should make any difference to the OPC. In principle the OPC is determining if a particle scatters or not. If it does, then it is counted. So even if RH affects scattering (which it will), then I do not see how it will stop the scattering all together such that a particle is not counted. The authors need to provide a plausible hypothesis at least to explain this issue. What does the manufacturer say the specifications should be for the OPC sensor? It seems like no attempt was made to contact the manufacturer to get an idea of how the mass is calculated. Given they are assessing their instrument; one would think they would be agreeable to helping them out. How do these results compare with what the manufacturer says it should do in terms of accuracy and precision?

**Response:**

The effect of RH and particle hygroscopicity upon particle refractivity and size is well known.  $\kappa$ -Kohler theory allows the effect to be modelled. Hygroscopic particles take up water as a function of RH, with more water taken up at higher RH. Typically, this effect is particularly important for inorganic aerosols. We explain this at the start of Section 3.3 (Page 20, line 3):

"Clearly there were times when there was a significant instrument artefact for the OPC-N2 (Figs 4 and S4) and the highest over-estimations occurred at high RH at both EROS and Tyburn Rd (e.g. Fig 5 and 6). The size of hygroscopic particles is known to be dependent on RH, as the particle refractive index and size are both a function of RH. Inorganic aerosols (e.g. sodium chloride, nitrate and sulphate), make up a large portion of the PM10 observed at EROS (Yin et al., 2010), and are known to demonstrate an exponential increase in hygroscopic growth at high RH (e.g. (Hu et al., 2010; Pope et al., 2010)."

We also note that Section 3.4 (Page 25, line 16) is a discussion on the cause of the OPC-N2 interference, and in this section we directly attribute this artefact to particle water content, as we stated in at page 21, starting at line 4:

"In the previous sections, the significant positive artefact observed by the OPC-N2 relative to the reference instruments were at times when the ambient RH was high, pointing to particle water content as the cause. This result is perhaps not surprising, as many studies in the literature have shown that particle water content can be a major reason for discrepancies between techniques that measure ambient particle mass (See e.g. (Charron et al., 2004)). The use of  $\kappa$ -Kohler theory to derive a correction factor based on ambient RH improved the agreement between the OPC-N2 and reference instruments"

Therefore, this artefact due to RH is not whether or not a particle is counted, rather the size bin that the particle is assigned to. Thus, as the OPC-N2 on-board calculation applies a single particle density for all size bins to convert the particle number concentration to particle mass, assigning a particle to wrong size bin will result in an over-estimation of the particle mass concentration.

We did contact Alphasense for more information on how the particle mass was calculated but they were unwilling to share that information with us, which was also the experience of Sousan et al. (2016). The manual of the OPC-N2 does not give any information with regards to accuracy and precision of the calculated particle mass concentrations, only for the number size distributions. This was part of the reason for focusing on particle mass concentrations.

8. There are many studies where mobile measurements of PM were made in urban and suburban areas. By looking at the spatial variation of the PM in those studies, one can get an idea of what kind of inter-instrument variability is required for this to be a useful instrument. Some attempt at this should be done, at least qualitatively.

**Response:**

The spatial variability of  $PM_{10}$  mass concentrations in urban areas is hugely variable, ranging from limited (e.g. 20-24 µg m-3 (Harrison et al., 1999), to more substantial such as 24-40 µg m-3 (Boogaard et al., 2010), 67-142 µg m-3(Chan et al., 2001), and likely reflects the spatial heterogeneity of the major sources (e.g. traffic). Similar trends are also found for  $PM_{2.5}$  with one study finding the concentration ranged from 6.7-48.3 µg m-3 across a city (Martuzevicius et al., 2004).

Considering the CV reported for  $PM_{10}$  mass concentrations by 14 OPC-N2 (22±13%), then we would expect these instruments to be suitable precision for the many urban areas where there is notable spatial variation.

**Minor issues:**

Pg 2, line 2: the term "reasonable" is used here and not justified.

Please see our response to comment #1.

Pg 2, line 30: this line is awkwardly written. Remove the "are" and use "companies" or "manufacturers" but not both.

**Response:**

Changed to:

"There are a wide range of low-cost particle sensors available commercially from manufacturers including Dylos, TSI, Airsense and Alphasense."

Pg 3 , line 20: define "PUWP" and "dylos" Pg 3, line 19: add "the" before "dylos" (if I am reading this correctly)

Response:

Dylos is the name of the instrument, so does not need defining. The PUWP is an acronym and so the definition has now been included. The sentence now reads:

"Previous field testing of low-cost particle sensors has found that the Dylos (Steinle et al., 2015) and (Gao et al., 2015) performed well for ambient sampling of particle mass concentration in both an urban and rural environments when compared to reference instruments, however they were assessed were over a short period (4-5 days)."

Pg 3, line 21: remove the "s" from "environments" Pg 3, line 22: add "they" after "however" Pg 3, line 29: "sites" to "site" Pg 4, line 11: replace "were" with "used"

Response:

All of the above have been fixed

Pg 4, line 15-17: awkwardly written. Please improve. And remove "s" from "systems"

Response: Changed to:

"Therefore, we developed a custom built system for logging the OPC-N2 during the intercomparison, utilizing either a Raspberry Pi 3 or Arduino system."

Pg 5, line 17-18: it is not clear what this is supposed to be used for in this paper.

Response:

We collected RH data from the nearby met station. This has been added to the text:

"In addition, RH measurements from the nearby Elms Road Meteorological station were also obtained, which is located approximately 100 m away from EROS."

Pg 5, line 29: briefly describe what the point of the "filter dynamic system" is.

Response:

The following text has been added to explain the use of the FDMS

"the TEOM monitor was fitted with a Filter Dynamic Measurement System (FDMS) (Grover et al., 2006), to correct semi-volatile particle loss."

Pg 6, line 6: add an "s" to "OPC" Pg 8, line 15: awkwardly written. Please improve.

Response: Changed to:

"This demonstrates that the highest and lowest reporting OPC was not consistently reporting the highest and lowest PM2.5 concentration, respectively over the whole 3 day period."

Pg 9, line 20: far too speculative without backing it up.

Response:

Please see our response to comment #1

Pg 10, line 5: define what "consistent" means to you. Fig 3 indicates it is not at all consistent:

**Response:**

We have changed the text to include the mean and standard deviation as below:

"Throughout the measurement period, the CV was fairly consistent (mean of 0.22±0.13), with spikes in CV values evident during periods of high PM2.5 concentrations, in agreement with trends observed in Fig 1."

Pg 10, line 7: again, "reasonable" is too subjective. Pg 10, line 23: again, the use of "reasonable": : :..what does this mean?

**Response:**

Please see our response to comment #1

Pg 11, line 5: it should not agree with the GRIMM as you have already stated it is 20% off to begin with.

**Response:**

The reviewer makes a valid point and we have changed the text to read:

"While the TSI and GRIMM have the same particle size cut-off (0.3  $\mu$ m), these instruments have been shown to disagree (Fig S1) possibly due to different particle collection efficiencies."

Pg 15: how is the volatile fraction determined? (briefly). What does "gravimetrically corrected" mean in this context?

**Response:**

The volatile fraction is determined by the FDMS system on the TEOM, and represents the mass of semi-volatile particles. We have added an explanation to the caption on Fig 7:

"Figure 7: Time series for hourly measured PM mass concentrations by the TEOM, four OPC-N2 and GRIMM at Tyburn Rd urban background AURN station. The volatile particle mass concentration as measured by the TEOM-FDMS and relative humidity measured at Tyburn Rd also shown."

The term gravimetrically corrected means that the optical instruments have been corrected by comparison to gravimetric determination of particle mass.

Table 2: units of slope? Or unitless?

Response:

The slopes are unit less as we have plotted measurements of the same units.

Pg 17, line 1: is this the median of all OPCs or all them individually?

**Response:**

Each OPC-N2 at Tyburn Rd was plotted as function of RH and showed the same trends.

Pg 21, lines 7-8: this has no bearing on the current study.

**Response:**

We disagree, this statement is entirely relevant to the current work as we have found that RH was a major artefact on the measured particle mass concentrations by the OPC-N2. This statement shows that this artefact due to particle water content is not just specific to the OPC-N2 but generally an issue across instruments that measure particle mass concentrations.

Pg 22, line 15: what is "knock on"??

*Response: We have removed this term.*

Pg 22, line 20: remove "while"

Fixed

Pg 22, line 23: "suitable" is not what the reader gets from this paper. See my comments above.

**Response:**

We disagree, as we have stated in our response to previous comments (#2) and will keep this sentence the same

Figure 1: difficult to see as there are too many lines. Perhaps shorten the time scale and zoom in. Perhaps a log scale would help too.

Figure 5: too small to see anything other than the peak. Perhaps use a log scale to better see what is going on.

Figure 6: Too small to see anything. I suggest you split the y-axis and zoom in to where the majority of data is.

**Response:**

Figures 1 and 5 have been fixed as suggested.

Figure 6: We have not split the y axis as suggested as we want to show all the data, the point of this figure is to show times when the OPC-N2 over-estimated the PM concentration, and splitting the y-axis would lose this information.

**Response to interactive comment from W.R. Stanley**

1. Albeit briefly, European Standard EN481 is mentioned in the OPC-N2 user manual when describing how PM is calculated from the particle number concentration data.

**Response:**

We have added that particle mass concentrations are calculated by OPC-N2 according to EN481 to the Section 2.1.1, please see response to Reviewer 2, comment #4.

2. The author could be more specific about the inlet arrangements with their use of the OPC-N2. In addition to comments made in this subject by referee RC2, with its small fan aspirator, the air-flow through the device may easily affected by changes to its default inlet or the nature of the ambient air e.g. breeze across the inlet. Possible differences in response between these and the reference instruments due to such factors should be discussed.

**Response:**

Please see our response to Reviewer 2, comment #6 on this issue. We have added discussion that the inlet arrangement may have affected the inter-comparison.

**References:**

Boogaard, H., Montagne, D.R., Brandenburg, A.P., Meliefste, K., Hoek, G., 2010. Comparison of short-term exposure to particle number, PM10 and soot concentrations on three (sub) urban locations. Science of The Total Environment 408, 4403-4411.

Castellini, S., Moroni, B., Cappelletti, D., 2014. PMetro: Measurement of urban aerosols on a mobile platform. Measurement 49, 99-106.

Chan, L.Y., Kwok, W.S., Lee, S.C., Chan, C.Y., 2001. Spatial variation of mass concentration of roadside suspended particulate matter in metropolitan Hong Kong. Atmospheric Environment 35, 3167-3176. Charron, A., Harrison, R.M., Moorcroft, S., Booker, J., 2004. Quantitative interpretation of divergence between PM10 and PM2.5 mass measurement by TEOM and gravimetric (Partisol) instruments. Atmospheric Environment 38, 415-423.

Dinoi, A., Donateo, A., Belosi, F., Conte, M., Contini, D., 2017. Comparison of atmospheric particle concentration measurements using different optical detectors: Potentiality and limits for air quality applications. Measurement 106, 274-282.

Gysel, M., Crosier, J., Topping, D.O., Whitehead, J.D., Bower, K.N., Cubison, M.J., Williams, P.I., Flynn, M.J., McFiggans, G.B., Coe, H., 2007. Closure study between chemical composition and hygroscopic growth of aerosol particles during TORCH2. Atmos. Chem. Phys. 7, 6131-6144.

Harrison, R.M., Jones, M., Collins, G., 1999. Measurements of the physical properties of particles in the urban atmosphere. Atmospheric Environment 33, 309-321.

Lewis, A.C., Lee, J.D., Edwards, P.M., Shaw, M.D., Evans, M.J., Moller, S.J., Smith, K.R., Buckley, J.W., Ellis, M., Gillot, S.R., White, A., 2016. Evaluating the performance of low cost chemical sensors for air pollution research. Faraday Discussions 189, 85-103.

Manikonda, A., Zíková, N., Hopke, P.K., Ferro, A.R., 2016. Laboratory assessment of low-cost PM monitors. Journal of Aerosol Science 102, 29-40.

Martuzevicius, D., Grinshpun, S.A., Reponen, T., Górny, R.L., Shukla, R., Lockey, J., Hu, S., McDonald, R., Biswas, P., Kliucininkas, L., LeMasters, G., 2004. Spatial and temporal variations of PM2.5 concentration and composition throughout an urban area with high freeway density—the Greater Cincinnati study. Atmospheric Environment 38, 1091-1105.

Sousan, S., Koehler, K., Hallett, L., Peters, T.M., 2016. Evaluation of the Alphasense optical particle counter (OPC-N2) and the Grimm portable aerosol spectrometer (PAS-1.108). Aerosol Science and Technology 50, 1352-1365.

Von der Weiden, S., Drewnick, F., Borrmann, S., 2009. Particle Loss Calculator–a new software tool for the assessment of the performance of aerosol inlet systems. Atmos. Meas. Tech 2, 479-494. Wang, Y., Li, J., Jing, H., Zhang, Q., Jiang, J., Biswas, P., 2015. Laboratory Evaluation and Calibration of Three Low-Cost Particle Sensors for Particulate Matter Measurement. Aerosol Science and Technology 49, 1063-1077.

Wang, Z., Calderón, L., Patton, A.P., Sorensen Allacci, M., Senick, J., Wener, R., Andrews, C.J. and Mainelis, G., 2016. Comparison of real-time instruments and gravimetric method when measuring particulate matter in a residential building. *Journal of the Air & Waste Management Association*, *66*, pp.1109-1120.

**Evaluation of a low-cost optical particle counter (Alphasense OPC-N2) for ambient air monitoring**

- 3
- Leigh R. Crilley1, Marvin Shaw2, Ryan Pound2, Louisa J. Kramer1, Robin Price3, Stuart
  Young2, Alastair C Lewis2, Francis D. Pope1\*
- 6

1School of Geography, Earth and Environmental Sciences, University of Birmingham,
8 Birmingham, United Kingdom, B15 2TT

---

## Author Response (AR2)

**Dear Editor,**

We thank you for the time and care you have put into this paper. We also agree that this paper is an important one and so have carefully addressed all of your comments, and modified the text accordingly. Please see our responses below.

**1.** While I appreciate the work done for improving the manuscript based on reviewers suggestions/comments, I still feel some additional corrections are needed before publication.

Essentially, based on the evidence provided, I am not fully convinced that "the OPC-N2 instrument demonstrated reasonable agreement with reference instrument for the measured mass concentration", which seems like a very positive statement considering the obvious limitations for its applicability. I believe you have to be much more cautious in your conclusions.

**Response:**

We think that for a low-cost sensor that the OPC-N2 does show reasonable agreement to the reference instruments used in this paper, especially considering that the OPC-N2 is at least 1/100th the cost of a TEOM-FDMS system. We are not trying to say that the agreement was excellent or great but rather can be considered as reasonable for low-cost sensor as evidenced by the stated precision and accuracy throughout the paper. The precision of the OPC-N2 was found to be unaffected by RH, with a CV of 22+13% between the 14 OPC-N2 employed in this study for PM10. The accuracy of the OPC-N2 relative to the optical reference instruments, as shown by the similar distributions in measured particle mass concentrations (Fig 4), suggests that there was similar measurement. A large positive artefact was observed at high RH (>85%), hence the need for a correction factor, which resulted in OPC-N2 being within 33% of a TEOM-FDMS. Therefore, we believe we are justified in stating, for a low-cost sensor, the agreement between reference and OPC-N2 measurements was reasonable.

*Therefore to clarify this point we have modified this sentence found in the abstract (line 23) and Section 4.0 (line 22) to now read:*

**"Comparison of the OPC-N2 to the reference optical instruments demonstrated reasonable agreement, for a low-cost sensor, to the measured mass concentrations of PM1, PM2.5 and PM10."**

**2.** First-of-all, there is a clear need for RH correction and this is far from being obvious. Humidograms in Fig. 8 are biased with respect to usual hygroscopic growth curves showing an increase from RH85% instead of RH70% for natural aerosol – your statement that « at RH <85% the hygroscopic growth of real aerosol is small » is actually false and should be RH70%-. I believe this is because you are considering ambient RH while the RH inside the instrument after travelling through the inlet is already lower. This can be fixed for the specific case of this study but without internal RH measurement, the correction may be quite different. This means, the instrument cannot be used without a clear RH controlling process. Then, you apply a correction assuming a  $\kappa$ -value. In practice, given the short duration of your experiment, a constant  $\kappa$  is acceptable, but it is no longer the case when instrument is used for longer periods. I believe you actually can see different aerosol populations in fig8 at high RH, leading to huge differences in OPC/TEOM. Applicability of a constant RH/  $\kappa$  correction scheme is to be questioned for longer measurement periods. This has to be made more clear.

Response: In the paper, we state that "The ratio of measured mass concentrations by the OPC-N2 relative to the reference instruments was plotted as a function of RH, and appeared to show an exponential increase above ~85% RH, similar to hygroscopic particle growth curves."

**And**

"Typically, at RH <85% the hygroscopic growth of real atmospheric aerosols is small and it may be more appropriate to apply a linear regression correction factor for data recorded under these RH conditions"

Thus we were not saying that there would be no hygroscopic growth under 85% RH, rather that we observed a near exponential increase above 85%, and that below this threshold a linear regression may be more appropriate.

The correction factor developed requires the input of ambient RH because we believe the measured PM mass is due to the hygroscopic growth of the particles. We agree that a calibration that uses internal RH would be better but this is impossible because the OPC-N2 cannot be opened. We did consider using a further calibration parameter that accounted for the change from ambient to internal RH. However, we felt that incorporating multiple factors in the calibration led to less confidence in the calibration because multiple parameters always lead to a better fit whether or not they are physically realistic. We also note that since the OPC-N2 inlet does not have a sheath flow gas then the temperature difference, and hence RH difference, between ambient and inside the instrument is likely small. To clarify this potential issue with the correction factor, we have included the following text at line 33, page 18:

"Ideally, a measure of RH internal to the instrument could be made to allow for calculation of particle hygroscopicity within the instrument. However, the OPC-N2 design does not allow for this, so we assume that ambient and instrument RH are identical. In reality, the instrument is likely to be slightly warmer than ambient and hence the RH within the instrument will be slightly lower than ambient. This difference will result in a lower apparent hygroscopicity."

We do not agree with the statement the OPC cannot be used without clear RH controlling process, as we show that below an RH of 85% the correction factor was near linear. This suggests that what is needed is calibration, an issue common to all instruments.

We note that we do discuss the issue of the large spread at high RH in Fig 8 in Section 3.4, where we discuss that this is likely due to changes in aerosol composition and that this would affect the  $\kappa$ -value, we already state the k is composition dependent. Therefore we have added the following in Section 3.4 to discuss the implications on long-term monitoring:

"Variation in ambient particle composition could account for the large spread observed in the ratio of OPC-N2/TEOM at high RH (Fig 8), as an average hygroscopicity correction will overestimate when PM with higher hygroscopicity is measured and vice versa for lower hygroscopicity particles. This would have potentially significant implications when using the OPC-N2 for longer-term monitoring, as the  $\kappa$  value may not be constant over the monitoring period. Therefore, this suggests the need for regular calibrations to account for changes in bulk aerosol composition and as a result  $\kappa$  values."

To emphasize this important point that  $\kappa$ -value is composition dependent to the reader; we have added text to the conclusions, see our response to comment 4.

**3.** I am also questioning Table 3 and the huge difference for TEOM in PM2.5 and PM10, which does not seem to be discussed. There is a loss mechanism somewhere that is not accounted for it seems.

**Response:**

We have added the following text as a discussion on the difference observed for PM2.5 and PM10 relative to the TEOM, at page 20, line 14:

"However, it was also evident from Table 3 that the slope was different for  $PM_{10}$  and  $PM_{2.5}$  mass fractions for all OPC-N2 when compared to the TEOM, and suggests that differing responses for the OPC-N2 to two size fractions. This may be related to the observed variation in  $\kappa$  between the size fractions relative to the TEOM or an unaccounted loss mechanism; the exact cause will be investigated further in future work."

**4.** Your paper is an important one and this is why you have to be very careful with the final statements. Essentially, what needs to be known is whether the instrument responds to PM quality objectives for air quality studies, and, if not, which kind of corrections are needed for its applicability. For me, your study shows that the sensor does not respond to DOQ as formulated for PM2.5 or PM10 standard application but, applying specific correction procedures, you can reach a reasonable agreement. Not vice versa. In the case of PM2.5 where the standard is a 1-yr average, I also tend to believe the correction procedures will be meaningless, given the intrinsic variability in RH/  $\kappa$ . This has to be made very clear in the conclusions, otherwise, the risk is high that your paper will be simply referenced as blank check for applying this sensor to any kind of PM studies, without the proper limitations in mind.

**Response:**

We have added the following text to Section 4.0 to highlight these limitations:

"All low cost PM sensors will likely require calibration factors to obtain the dry particle weight unless they actively dry the PM containing air stream before it enters the device. The use of heated inlets could be used to reduce the RH in the air stream but would have consequences on the power requirements of the sensor, potentially making them less attractive for battery led operation. Thus it highlights that the OPC-N2 does not respond the same as reference instruments to ambient particle mass, but provided appropriate correction factors are applied, reasonable agreement with OPC-N2 to reference instruments can be achieved. Furthermore, the dependence of the OPC-N2 on a correction for RH and  $\kappa$  may limit its application for longer-term monitoring as the  $\kappa$  value may change over time, and this will be the focus of future work. This is especially salient when considering using the OPC-N2 to compare to air quality standards that are one-year averages of PM2.5 and PM10."

In conclusion, I expect that you modify the paper in this direction before proceeding with publication.

**Evaluation of a low-cost optical particle counter (Alphasense OPC-N2) for ambient air monitoring**

- Leigh R. Crilley1, Marvin Shaw2, Ryan Pound2, Louisa J. Kramer1, Robin Price3, Stuart
  Young2, Alastair C Lewis2, Francis D. Pope1\*
- 6

[revised manuscript text omitted]

---

## Author Response (AR3)

*Dear Editor,*

*We thank you again for your time and attention to the paper. We have modified the abstract as per your request.*

*Yours Sincerely,*

*Francis Pope*

---

## Author Response (AR4)

*Dear Editor,*

*We thank you again for your time and attention to the paper. We have modified the abstract as per your request.*

*Yours Sincerely,*

*Francis Pope*